# Unveiling Super Experts in Mixture-of-Experts Large Language Models

**Zunhai Su[1], Qingyuan Li[2], Hao Zhang[2], Weihao Ye[3], Qibo Xue[4], Yulei Qian[2], Yuchen Xie[2], Ngai Wong[5], Kehong Yuan[1]**

[1]Tsinghua University  [2]Meituan  [3]Xiamen University  [4]Nanjing University
[5]The University of Hong Kong

## Abstract

Leveraging the intrinsic importance differences among experts, recent research has explored expert-level compression techniques to enhance the efficiency of Mixture-of-Experts (MoE) large language models (LLMs). However, existing approaches often rely on empirical heuristics to identify critical experts, while lacking a deeper understanding into the heterogeneous importance of experts and the inner workings of MoE LLMs. In this study, we report, for the first time, the discovery and systematic investigation of a distinct subset of experts that play a pivotal role in the model's forward inference. These experts are prevalent in open-source MoE LLMs, and despite their extremely limited number, pruning them results in a substantial decline in model performance (e.g., prune just three out of 6,144 causes Qwen3-30B-A3B to generate repetitive and uninformative outputs). We refer to these experts as *Super Experts (SEs)*. Our comprehensive analysis provides progressively deeper insights into SEs: *(i)* SEs are characterized by rare but extreme activation outliers in the output of the `down_proj`, which give rise to massive activations in the hidden states between decoder layers. Moreover, the distribution of SEs is model-specific, data-agnostic, and remains unaffected by post-training processes. *(ii)* By pruning SEs, we assess their significance across a variety of tasks, revealing their considerable impact on the model's overall performance, particularly in mathematical reasoning. *(iii)* We further investigate why compressing SEs exerts such a pronounced impact. We show that, in MoE LLMs, SEs serve as the primary source of the systematic outlier mechanism in Transformers, and that compressing them profoundly disrupts this process, ultimately causing the collapse of attention sinks. These findings advance the understanding of the internal dynamics of MoE LLMs, filling an important gap in the current knowledge. In addition, we developed an automated tool for rapid and accurate SE profiling. The code is provided in https://github.com/ZunhaiSu/Super-Experts-Profilling.

## 1 Introduction

Sparsely activated Mixture-of-Experts (MoE) models employ dynamic routing and sparse activation, demonstrating significant potential in enhancing the learning capacity of large language models (LLMs) (Cai et al., 2024; Mu & Lin, 2025). This paradigm has led to the development of state-of-the-art MoE LLMs, including DeepSeek (Guo et al., 2025; Liu et al., 2024), Qwen (Yang et al., 2025a), LongCat-Flash (Team et al., 2025b;a) and others. Despite their potential, a significant challenge stems from their large parameter size and high computational cost (Li et al., 2023; Lu et al., 2024; Chowdhury et al., 2024), which present considerable obstacles for deployment. Model compression techniques, such as quantization (Frantar et al., 2022; Xiao et al., 2023a; Su et al., 2025a), pruning (Frantar & Alistarh, 2023; Sun et al., 2023) and others (Zhu et al., 2024; Wang et al., 2024), enable the development of more compact and computationally efficient models.

Beyond LLM-oriented compression approaches, expert-level compression methods have been developed by leveraging the structural characteristics of MoE models and the uneven importance of experts induced by training strategies (Chowdhury et al., 2024; Chi et al., 2022; Lu et al., 2024).

Specifically, it employs various expert importance metrics to guide the pruning, merging, or skipping of less critical experts (Lu et al., 2024; Huang et al., 2025; Xie et al., 2024), prioritize more important ones by assigning higher bit budgets during quantization (Duanmu et al., 2025; Li et al., 2024), and allocate more ranks in low-rank decomposition (Yang et al., 2024; Li et al., 2023). For instance, several works evaluate expert importance by measuring activation frequency or by analyzing router scores produced within MoE layers (Li et al., 2024; 2023; Huang et al., 2025). Additionally, reconstruction loss and other similarity-based metrics have been utilized in previous studies (Lu et al., 2024; Duanmu et al., 2025; Zhang et al., 2024).

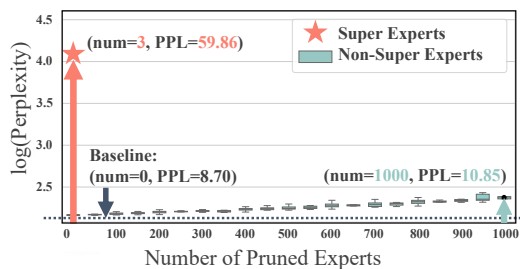

Figure 1: Analysis of experts pruning on Qwen3-30B-A3B using the WikiText-2 dataset. Pruning three Super Experts results in a significant degradation in Perplexity (PPL).

Analyzing expert importance not only facilitates model compression but also provides deeper insights into the inner workings of MoE LLMs Zhang et al. (2026). However, existing approaches often rely on empirical criteria to identify critical experts, lacking a deeper exploration and understanding of the heterogeneous importance among experts. In this study, we address a fundamental yet previously overlooked question: ***Is there a small subset of distinct experts that plays an exceptionally critical role in the underlying mechanisms of MoE LLMs?***

Through comprehensive analysis of various open-source MoE LLMs, we consistently confirm the existence of such experts. Despite their extreme limited number, pruning these experts leads to a significant collapse in model performance. As shown in Figure 1, pruning just three experts from Qwen3-30B-A3B leads to a significant degradation in model performance, while randomly pruning other experts results in a considerably smaller impact. We refer to these experts as Super Experts (SEs), and our comprehensive analysis provides progressively deeper insights into SEs.

In Section 3, we first characterize SEs and analyze their distribution across various models and input data domains. SEs are identified by extreme activation outliers in the output of the `down_proj`, which induce massive activations (MAs) (Sun et al., 2024). Intriguingly, the distribution of SEs remains model-specific, data-agnostic, and the SEs in the base model maintain consistency after post-training processes. In Section 4, we assess the importance of SEs by quantifying performance degradation following their dynamic pruning. Notably, pruning SEs leads to a complete performance collapse with Pass@1 dropping to nearly zero on tasks such as AIME and Math-500 (AIME, 2024; 2025; Lightman et al., 2023) for reasoning LLMs. In Section 5, we further deepen our understanding of SEs by revealing their origin in the behavior of systematic outliers mechanism in Transformers (Su & Yuan, 2025; An et al., 2025). Our findings confirm that MoE LLMs rely on SEs to induce attention sinks, which are crucial for the distribution of attention scores and must be preserved during sparse attention or KV compression (Xiao et al., 2023b; Su et al., 2025a).

The main contributions of this work are summarized as follows:

• We provide the first comprehensive characterization of Super Experts (SEs) in MoE LLMs, an exceptionally rare yet fundamentally critical subset of experts, thereby filling a significant gap in the existing understanding of MoE LLMs. Extensive analyses across multiple models and tasks reveal key properties of SEs, including their stable distribution and critical impact on model performance.

• We demonstrate that SEs are the primary drivers of systematic outliers in Transformers. In MoE LLMs, their strong activation on attention sink tokens makes them the fundamental source of these outliers, and compressing them severely disrupts this process, ultimately leading to the collapse of attention sinks.

• Our findings on SEs provide new insights into the internal dynamics of MoE LLMs and the heterogeneous importance of experts. These insights serve as a foundation for designing more expert-balanced pre-training regimes and for advancing robust expert compression strategies.

## 2 PRELIMINARIES ON MoE LLMs

**MoE LLMs.** LLMs are typically structured as a stack of Transformer decoder blocks (Vaswani et al., 2017), each consisting of a multi-head self-attention (MHSA) layer and a feed-forward network (FFN) layer. In MoE LLMs, the FFN layers are replaced by MoE layers, where each layer consists of multiple experts, each represented by a FFN. A concise overview of MoE LLMs is presented in Figure 2. Let $H^0 \in \mathbb{R}^{n \times d}$ represent the input to the first decoder, where $d$ is the embedding dimension, and $n$ is the length of the tokenized input sequence. Then, the output of the $l$-th decoder block, $H^l \in \mathbb{R}^{n \times d}$, is given by:

$$H^l = \text{MoE}\left(\text{LN}_{moe}\left(H^{l'}\right)\right) + H^{l'}, \tag{1}$$

$$H^{l'} = O^l + H^{l-1}, O^l = \text{MHSA}\left(\text{LN}_{mhsa}\left(H^{l-1}\right)\right), \tag{2}$$

where $1 \leq l \leq L$, with $L$ denoting the total number of blocks. LN refers to layer normalization, $O^l$ representing the output of the MHSA, and $H^{l'}$ denoting residual summations after the MHSA.

**MoE Layer.** The hidden representation after MHSA, $H^{l'}$, passes through a LN and then fed into the MoE layer. First, the router network determines which experts to activate and how to scale their outputs through the weight matrix $W_G$. The routing weights $G \in \mathbb{R}^{n \times E}$ are computed as:

$$G = \text{softmax}(H^{l'} W_G). \tag{3}$$

Figure 2: Decoder Architecture of MoE LLM.

Then, sparse activation of the experts is achieved by selecting the top-$k$ routing weights for each input token. The output of the activated experts is scaled by the routing weights and aggregated to form the output of the MoE layer:

$$\sum_{i \in \text{Top-}k(G_j)} G_{ji} \cdot \text{FFN}\left(\text{LN}_{moe}(H_j^{l'})\right), \quad \forall j = 1 \ldots n, \tag{4}$$

where Top-$k(G_j)$ denotes the indices of the top-$k$ routing weights for the $j$-th input token. The FFN is defined as:

$$\text{FFN}\left(X\right) = \left(\sigma\left(X W_g\right) \odot X W_u\right) W_d, \tag{5}$$

where $W_g$, $W_u$, and $W_d$ are the weight matrices for the gating, up-projection, and down-projection, respectively. $\sigma$ denotes the activation function, and $\odot$ represents the Hadamard product.

## 3 SUPER EXPERTS: DISCOVERY AND LOCALIZATION

In this section, we first demonstrate the discovery process of SEs using Qwen3-30B-A3B as an example. Next, we analyze SEs across different MoE LLMs and data domains to examine their distribution patterns and highlight the widespread presence of SEs.

### 3.1 SUPER EXPERTS INDUCE MASSIVE ACTIVATIONS

Recent research has explored a distinct class of extreme activation outliers in LLMs, which appear in the hidden states between decoder layers and are known as massive activations (MAs) (Sun et al., 2024; Guo et al., 2024). They are limited in number, yet their values are orders of magnitude larger than those of other activations (e.g., up to 100,000 times larger). The discovery of SEs arises from an exploration and analysis of the formation of MAs (Sun et al., 2024) in MoE LLMs. Existing research has yet to clarify how these MAs arise in MoE LLMs. Do these activations arise from the collective activity of all activated experts, are they primarily driven by some specific experts, or are they instead caused by other components of the model?

Through analysis of several prominent open-source MoE LLMs (e.g., Qwen series, DeepSeek series, Mistral), we surprisingly find that a small subset of experts consistently produces extreme activation

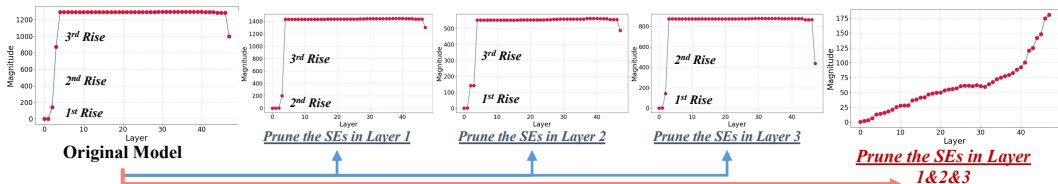

Figure 3: SEs mechanism in Qwen3-30B-A3B. The line plots depict the maximum output magnitudes of `down_proj` for experts 68/92/82 across layers. Massive activation is gradually amplified through expert 68 in layer 1, expert 92 in layer 2, and expert 82 in layer 3. Extreme activation outliers from these SEs are propagated into the hidden states between decoders via residual summation, progressively leading to massive activation.

Figure 4: Impact of SEs pruning on MAs in Qwen3-30B-A3B. MAs are computed using 100 input samples from the C4 (Raffel et al., 2020) dataset, each with a length of 2K.

outliers in the output of their `down_proj` layers. These outliers are subsequently passed onto the hidden states via residual summation after the MoE layers, leading to MAs. The entire process is illustrated in Figure 3 using Qwen3-30B-A3B as example. This phenomenon typically occurs in a single layer (e.g., Mixtral) or in just a few layers (e.g., Qwen3-30B-A3B) starting from the initial decoder layers, ultimately leading to stable MAs across nearly all subsequent layers. To directly validate this mechanism, we also perform ablation experiments by dynamic pruning the SEs in Qwen3-30B-A3B. As illustrated in Figure 4, pruning SEs from a single layer effectively eliminates their contribution to MAs. Furthermore, when all SEs are pruned, MAs are completely eliminated, confirming that they are directly generated by SEs.

### 3.2 LOCALIZATION OF SUPER EXPERTS

#### 3.2.1 SUPER EXPERTS PROFILING

Given that SEs are defined by their influence on the formation of MAs through the extreme activation outliers they generate, we propose the following broad yet effective quantitative definition. Specifically, we compute the maximum output magnitudes to the `down_proj` for all experts across all layers. Let $L$ denote the set of layers responsible for the formation of MAs. Let $a_{l,e}$ denote the maximum output magnitude to the `down_proj` of expert $e$ in layer $l$, and let $\mathcal{A} = \{a_{l,e}\}$ be the set of all such values across the entire model. An expert $e$ in layer $l$ is classified as a SE if:

$$a_{l,e} > P_{99.5} \quad \text{and} \quad a_{l,e} > \frac{1}{10}a_{\max} \quad \text{and} \quad l \in L \tag{6}$$

where $P_{99.5} = \text{Percentile}_{99.5}(\mathcal{A})$ and $a_{\max} = \max \mathcal{A}$. This criterion is motivated by the heavy-tailed distribution of $a_{l,e}$ and effectively identifies the experts of interest across various MoE LLMs, as highlighted in bold in Table 1. Additional analyses are provided in Appendix I. No specific dataset is designated for identifying SEs, since we later demonstrate that their distribution remains stable across different input datasets. The pseudocode of SEs profiling is presented in Appendix L. We have developed an automated tool for rapid and precise SE profiling based on this definition. The code is provided in https://github.com/ZunhaiSu/Super-Experts-Profiling.

#### 3.2.2 DISTRIBUTION OF SUPER EXPERTS ACROSS MODELS AND DATA DOMAINS

We select three representative MoE LLMs with distinct designs for analysis: Qwen3-30B-A3B, DeepSeek-V2-Lite-Chat, and Mixtral-8x7B-Instruct-v0.1. We also include the base

Table 1: Activations identified as SEs are highlighted in bold, based on results from the C4 dataset.

| Model | Total Experts | SE Count | SE Proportion | Top 1 | Top 2 | Top 3 | Top 4 | Top 5 | Top 10 | Top 0.5% | Top 1 * 0.1 |
|---|---|---|---|---|---|---|---|---|---|---|---|
| Qwen3-30B-A3B | 6144 | 3 | 0.05% | **744.0** | **540.0** | **430.0** | 63.5 | 19.1 | 12.1 | 7.3 | 74.4 |
| DeepSeek-R1 | 15677 | 10 | 0.06% | **616.0** | **536.0** | **171.0** | **143.0** | **143.0** | 67.0 | 36.75 | 61.6 |
| DeepSeek-V2-Lite-Chat | 1782 | 2 | 0.11% | **1424.0** | **462.0** | 112.5 | 89.5 | 37.5 | 24.0 | 34.5 | 142.4 |
| Mixtral-8x7B-Instruct-v0.1 | 256 | 1 | 0.39% | **5600.0** | 302.0 | 286.0 | 258.0 | 253.0 | 139.0 | 5600.0 | 560 |

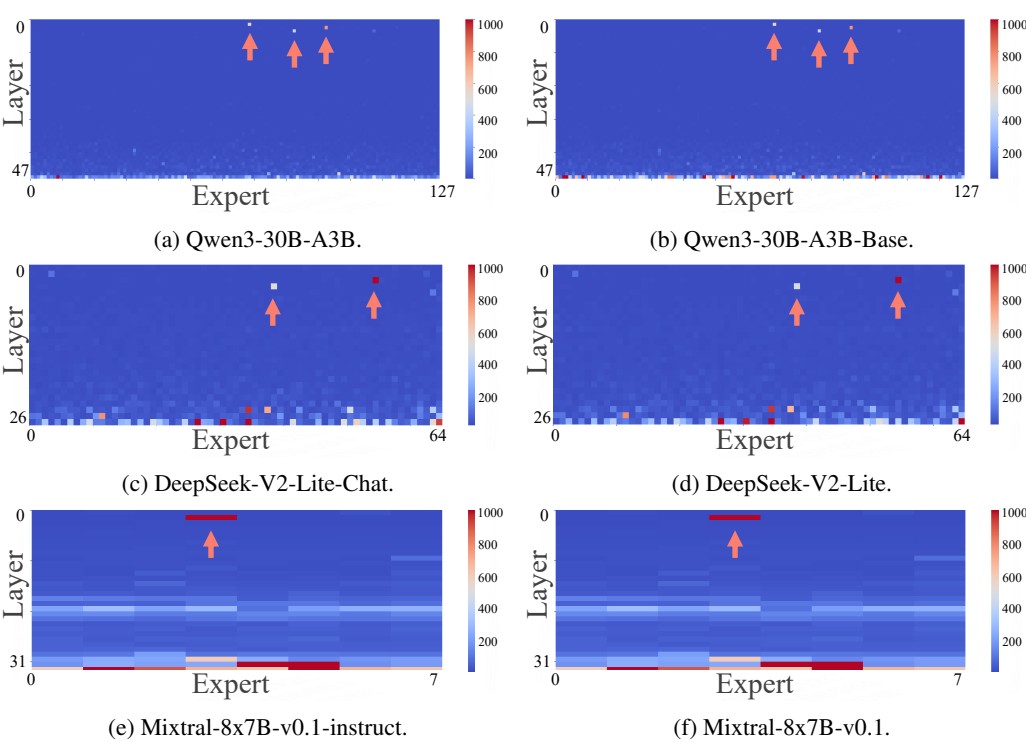

(a) Qwen3-30B-A3B.

(b) Qwen3-30B-A3B-Base.

(c) DeepSeek-V2-Lite-Chat.

(d) DeepSeek-V2-Lite.

(e) Mixtral-8x7B-v0.1-instruct.

(f) Mixtral-8x7B-v0.1.

Figure 5: Heatmap visualizations of the maximum output magnitudes from the `down_proj` for each expert across layers. SEs are highlighted with arrows.

models of these three LLMs to illustrate the impact of post-training processes. Although all of these models are MoE LLMs, they exhibit distinct design differences. For instance, Qwen3 and Mixtral do not employ shared experts, whereas DeepSeek does. DeepSeek-V2-Lite adopts a hybrid architecture, wherein the first layer utilizes dense MLPs, while the remaining layers are based on MoE blocks. Through the proposed SE profiling tool, we identify the SEs in these models using the C4 (Raffel et al., 2020) dataset. A summary of the SEs is provided in Table 2, and heatmap visualizations of the maximum output magnitudes from the `down_proj` are shown in

Table 2: SEs of several MoE LLMs.

| Model | Super Experts |
|---|---|
| Qwen3-30B-A3B Qwen3-30B-A3B-Base | Layer 1 Expert 68, Layer 2 Expert 92, Layer 3 Expert 82 |
| DeepSeek-V2-Lite-Chat DeepSeek-V2-Lite | Layer 3 Expert 54, Layer 4 Expert 38 |
| Mixtral-8x7B-Instruct-v0.1 Mixtral-8x7B-v0.1 | Layer 1 Expert 3 |

Figure 5. The key conclusions regarding SEs are summarized as follows: *(i)* SEs are consistently present across the investigated models, accounting for less than 0.5% of all experts. *(ii)* After post-training processes, the distribution of SEs remains unchanged compared to the base model. Additional results on the distribution of SEs across training stages is provided in Appendix K. Moreover, some experts in the final layers also exhibit extreme activation outliers. However, since they do not contribute to the formation of MAs, they do not hold the same level of significance as SEs. Additional results are available in Appendix C.

In addition to the C4 dataset, we also analyze SE distributions across several other datasets, including WikiText-2 (Merity et al., 2016), C-Eval (Huang et al., 2023), GSM8K (Cobbe et al., 2021), and

Table 3: Evaluation of the importance of SEs in non-reasoning models. The results of random pruning are obtained by averaging the performance over five runs.

| Model | Setting | Avg. | ARC-c | ARC-e | BoolQ | GSM8K | Hella Swag | MMLU | Open BookQA | PIQA | Wino Grande | Wiki PPL |
|---|---|---|---|---|---|---|---|---|---|---|---|---|
| Qwen3 30B-A3B | Baseline | 70.22 | 52.65 | 79.50 | 88.72 | 89.61 | 59.63 | 77.82 | 34.20 | 79.33 | 70.56 | 8.70 |
| | Prune SEs | 55.00 | 46.08 | 76.05 | 70.73 | 42.38 | 39.31 | 56.03 | 29.80 | 72.52 | 62.12 | 59.86 |
| | Drop Rate (%) | **21.68%** | 12.48% | 4.34% | 20.28% | **52.71%** | 34.08% | 28.00% | 12.87% | 8.58% | 11.96% | - |
| | Random | 70.36 | 52.73 | 79.46 | 88.59 | 89.84 | 59.50 | 77.84 | 34.00 | 79.76 | 71.51 | 8.71 |
| | Drop Rate (%) | -0.20% | -0.15% | 0.05% | 0.15% | -0.26% | 0.22% | -0.03% | 0.58% | -0.54% | -1.35% | - |
| | LC Random | 70.21 | 52.60 | 79.55 | 88.63 | 89.55 | 59.45 | 77.80 | 34.00 | 79.43 | 70.86 | 8.70 |
| | Drop Rate (%) | 0.01% | 0.09% | -0.06% | 0.10% | 0.07% | 0.30% | 0.03% | 0.58% | -0.13% | -0.43% | - |
| DeepSeek V2-Lite | Baseline | 60.27 | 46.59 | 78.37 | 79.79 | 37.83 | 58.75 | 55.03 | 34.60 | 80.30 | 71.19 | 6.31 |
| | Prune SEs | 43.90 | 29.27 | 54.92 | 68.62 | 9.78 | 43.72 | 41.77 | 21.00 | 68.28 | 57.7 | 10.75 |
| | Drop Rate (%) | **27.17%** | 37.18% | 29.92% | 14.00% | **74.15%** | 25.58% | 24.10% | 39.31% | 14.97% | 18.95% | - |
| | Random | 60.30 | 46.50 | 78.45 | 80.37 | 37.38 | 58.77 | 55.10 | 34.40 | 80.14 | 71.59 | 6.31 |
| | Drop Rate (%) | -0.05% | 0.19% | -0.10% | -0.73% | 1.19% | -0.03% | -0.13% | 0.58% | 0.20% | -0.56% | - |
| | LC Random | 60.18 | 46.69 | 78.21 | 79.83 | 37.22 | 58.71 | 55.05 | 34.40 | 80.24 | 71.26 | 6.32 |
| | Drop Rate (%) | 0.15% | -0.21% | 0.20% | -0.05% | 1.61% | 0.07% | -0.4% | 0.58% | 0.07% | -0.10% | - |
| Mixtral 8x7B-v0.1 | Baseline | 67.84 | 56.57 | 84.26 | 85.02 | 57.32 | 64.89 | 67.83 | 35.60 | 82.48 | 76.56 | 3.84 |
| | Prune SEs | 49.38 | 36.01 | 64.44 | 75.66 | 24.34 | 50.6 | 42.47 | 20.60 | 73.12 | 57.22 | 6.23 |
| | Drop Rate (%) | **27.21%** | 36.34% | 23.52% | 11.01% | **57.54%** | 22.02% | 37.39% | 42.13% | 11.35% | 25.26% | - |
| | Random | 67.82 | 56.57 | 84.09 | 85.23 | 58.15 | 64.92 | 68.08 | 35.00 | 82.21 | 76.16 | 3.86 |
| | Drop Rate (%) | 0.02% | 0.00% | 0.20% | -0.25% | -1.45% | -0.05% | -0.37% | 1.69% | 0.33% | 0.52% | - |
| | LC Random | 67.68 | 56.48 | 84.16 | 85.14 | 57.83 | 64.73 | 67.52 | 35.60 | 81.14 | 76.55 | 3.85 |
| | Drop Rate (%) | 0.24% | 0.16% | 0.12% | -0.14% | -0.89% | 0.25% | 0.46% | 0.00% | 1.62% | 0.01% | - |

Table 4: Evaluation of the importance of SEs in DeepSeek-R1.

| Model | Setting | Avg. | GPQA Diamond | Math-500 | AIME 2024 | AIME 2025 | LiveCodeBench | Wiki PPL |
|---|---|---|---|---|---|---|---|---|
| | | | Pass@1 | Pass@1 | Pass@1 | Pass@1 | Pass@1 | |
| DeepSeek-R1 | Baseline | 75.64 | 71.50 | 97.60 | 79.33 | 66.33 | 63.44 | 3.33 |
| | Prune SEs | 1.81 | 5.05 | 4.00 | 0.00 | 0.00 | 0.00 | 5.18 |
| | Drop Rate (%) | **97.61%** | **93.0%** | **95.9%** | **100%** | **100%** | **100%** | - |
| | Random Pruning | 75.53 | 72.63 | 98.00 | 77.67 | 67.00 | 62.37 | 3.35 |
| | Drop Rate (%) | 0.15% | -1.58% | -0.41% | 2.09% | -1.01% | 1.69% | - |
| | LC Random Pruning | 75.51 | 71.50 | 98.00 | 78.67 | 67.00 | 62.37 | 3.36 |
| | Drop Rate (%) | 0.17% | 0.00% | -0.41% | 0.83% | -1.01% | 1.96% | - |

HumanEval (Chen et al., 2021). As shown in Appendix D, the distribution of SEs remains highly stable, regardless of variations in the input data domain.

## 4 THE IMPORTANCE OF SUPER EXPERTS

In this section, we assess the importance of SEs by measuring the performance drop caused by dynamically pruning them (i.e., skipping the experts when selected by the router). We use the original model and results from random pruning of an equivalent number of experts as baselines. Random pruning is implemented in two ways: globally across all layers, or within the same layers as the SEs, which we refer to as layer-controlled (LC) random pruning. To more effectively evaluate the importance of SEs, we utilize distinct benchmark types for non-reasoning and reasoning models.

### 4.1 IMPACT ON NON-REASONING MODELS

For non-reasoning models, we select three models: the non-thinking mode of Qwen3-30B-A3B, DeepSeek-V2-Lite and Mixtral-8x7B-v0.1. We utilize the datasets listed below and conduct evaluations using lm-eval (Gao et al., 2024b), including ARC-challenge (ARC-c), ARC-easy (ARC-e) (Clark et al., 2018), BoolQ (Clark et al., 2019a), GSM8K (Cobbe et al., 2021), HellaSwag (Zellers et al., 2019), MMLU (Hendrycks et al., 2021), OpenBookQA (Mihaylov et al., 2018), PIQA (Bisk et al., 2020), and WinoGrande (Keisuke et al., 2019). As shown in Table 3, pruning only a few SEs leads to significant degradation across all tasks, with average accuracy dropping by 21.68% to 27.21%. In particular, for GSM8K, the degradation ranges from 52.71% to 74.15%. In contrast, random pruning has a negligible impact, underscoring the crucial role of SEs.

Table 5: Evaluation of the importance of SEs in Qwen3-30B-A3B.

| Model | Setting | Avg. | GPQA Diamond | Math-500 | AIME 2024 | AIME 2025 | HumanEval | Wiki PPL |
|-------|---------|------|--------------|----------|-----------|-----------|-----------|----------|
| | | | Pass@1 | Pass@1 | Pass@1 | Pass@1 | Pass@1 | |
| Qwen3-30B-A3B | Baseline | 69.37 | 61.62 | 88.00 | 80.00 | 73.33 | 43.90 | 8.70 |
| | Prune SEs | 4.02 | 18.69 | 1.40 | 0.00 | 0.00 | 0.00 | 59.86 |
| | Drop Rate (%) | **93.62%** | **69.7%** | **98.4%** | **100%** | **100%** | **100%** | - |
| | Random Pruning | 69.33 | 61.62 | 89.00 | 80.00 | 73.33 | 42.70 | 8.71 |
| | Drop Rate (%) | 0.06% | 0.00% | -1.10% | 0.00% | 0.00% | 2.7% | - |
| | LC Random Pruning | 68.97 | 61.62 | 88.00 | 80.00 | 73.33 | 41.90 | 8.72 |
| | Drop Rate (%) | 0.58% | 0.00% | 0.00% | 0.00% | 0.00% | 4.56% | - |

## 4.2 IMPACT ON REASONING MODELS

For evaluating the importance of SEs in reasoning models, we select DeepSeek-R1 and the thinking mode of Qwen3-30B-A3B. We select benchmarks more suitable for testing reasoning models and conduct evaluations based on the EvalScope (Team, 2024). The generation configurations align with the corresponding technical reports of the models. These benchmarks are: *(i)* **General Tasks:** We use GPQA-Diamond under a 5-shot setting. GPQA (Rein et al., 2024) is a challenging dataset of multiple-choice questions authored by domain-specific multidisciplinary experts. *(ii)* **Math & Text Reasoning:** To evaluate mathematical and logical reasoning skills, we use high-level math benchmarks, including MATH-500 (Lightman et al., 2023), AIME'24 (AIME, 2024), and AIME'25 (AIME, 2025). *(iii)* **Agent & Coding:** To test the model's proficiency in coding and agent-based tasks, we use LiveCodeBench (Jain et al., 2024) and HumanEval (Chen et al., 2021).

The results, presented in Tables 4 and 5, show that pruning the SEs causes a significant performance degradation, while random pruning has almost no impact. The Pass@1 scores for most tasks drop to zero, highlighting the critical role of SEs. During the review of model responses on the Math-500 benchmark, we made a striking observation: after pruning the SEs, the model consistently generated repetitive responses in nearly every test, continuing until it reached the maximum output length, as shown in Table 11 and 12. This behavior suggests that the model loses its ability to reason and solve problems entirely after SE pruning, with additional discussion on this part provided in Appendix H. More results are in Appendix E.

## 5 UNDERSTANDING THE IMPACT OF SUPER EXPERTS COMPRESSION

Why are SEs so critical to MoE LLMs? In this section, we first reveal SEs as the primary source of systematic outliers in MoE LLMs. Then, we examine how compressing SEs affects the attention mechanism, providing both an in-depth understanding and a quantitative analysis.

### 5.1 SUPER EXPERTS AS THE ORIGIN OF SYSTEMATIC OUTLIERS IN MOE LLMS

Previous studies (Su & Yuan, 2025; An et al., 2025) have shown that Transformer-based dense LLMs exhibit systematic outliers. These outliers appear in multiple forms, including weight outliers (also referred to as super weights (Yu et al., 2024)), activation outliers (encompassing both activation spikes and MAs (Yang et al., 2025b; Sun et al., 2024)), and attention outliers (commonly known as attention sinks (ASs) (Xiao et al., 2023b)). Importantly, such outliers emerge, stabilize, and vanish in a systematic fashion, and they are crucial to the model's overall performance (Yu et al., 2024; Sun et al., 2024; Xiao et al., 2023b). A detailed discussion of related work on systematic outliers in Transformers is provided in Appendix B.

Building on prior research and our own findings on SEs, we demonstrate that SEs constitute the fundamental source of systematic outliers in MoE LLMs. Specifically, using Qwen3-10B-A3B as an example, the router scores assigned to SEs for the first token (which also serves as the attention sink token) are exceptionally large, whereas for non-sink tokens the scores are more evenly distributed across experts, as shown in the visualization calibrated on the C4 dataset in Figure 6. This routing behavior of SEs ensures that the attention sink token is strongly activated at the SEs. Notably, this behavior is independent of the input dataset (see Appendix F), which also explains why SEs distributions are model-specific. The sink token subsequently produces activation outliers

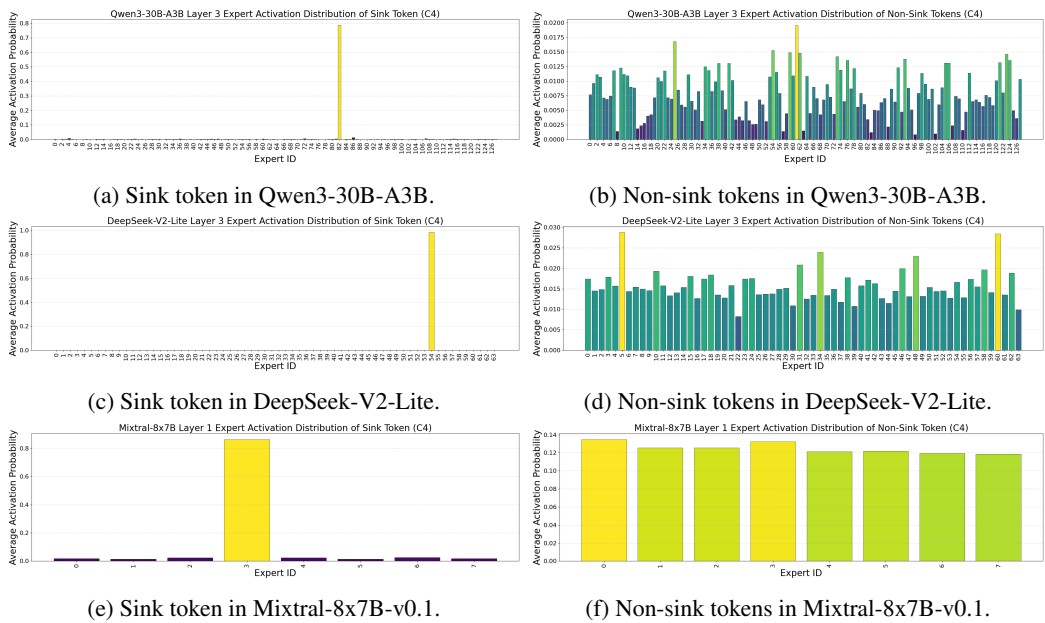

(a) Sink token in Qwen3-30B-A3B.  (b) Non-sink tokens in Qwen3-30B-A3B.

(c) Sink token in DeepSeek-V2-Lite.  (d) Non-sink tokens in DeepSeek-V2-Lite.

(e) Sink token in Mixtral-8x7B-v0.1.  (f) Non-sink tokens in Mixtral-8x7B-v0.1.

Figure 6: Expert router score distributions for sink and non-sink tokens, based on calibration using the C4 dataset. Additional experimental results are provided in Appendix F.

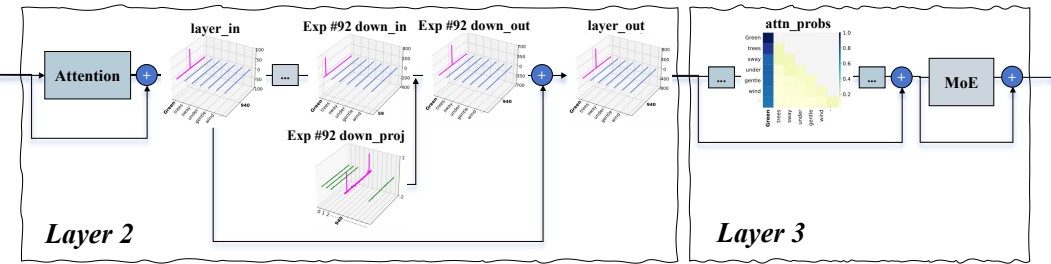

Figure 7: Systematic outlier mechanism in a single layer of Qwen3-30B-A3B, using the input: "Green trees sway under gentle wind." The complete illustration is provided in Figure 22.

in the output through weight outliers in the down_proj. Through the residual connections, these outliers propagate into the hidden states as MAs. At the attention layers, such tokens then attract disproportionate attention and ultimately emerge as attention sinks. Unlike dense LLMs, where such behavior typically occurs within a single layer (An et al., 2025; Su & Yuan, 2025), MoE models exhibit the progressive formation of systematic outliers by SEs across multiple layers. The overall process is illustrated in Figure 7. More detailed analyses of this process are presented in Appendix G. Additional cross-domain analyses of SEs and attention sink tokens are provided in Appendix J.

Within the dynamics of systematic outliers in MoE LLMs, *SEs constitute the primary source, MAs act as the intermediate bridge, and ASs manifest their effects within the attention mechanism.* This analysis underscores the pivotal role of SEs in the internal mechanisms of MoE LLMs and elucidates the distinctive manifestation of systematic outliers in Transformers within the MoE paradigm. Further weight-level experiments and interpretability analyses on SEs are presented in Appendix H.

## 5.2 SUPER EXPERTS COMPRESSION DISRUPTS ATTENTION SINKS

Given that SEs act as the primary source of systematic outliers and ASs embody their final manifestation, we posit that compressing SEs interferes with AS formation, thereby causing significant deterioration in model performance. StreamLLM (Xiao et al., 2023b) identified ASs in LLMs, in

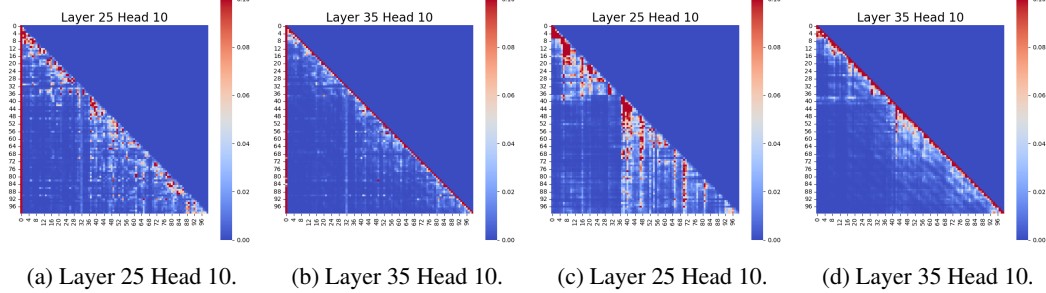

| (a) Layer 25 Head 10. | (b) Layer 35 Head 10. | (c) Layer 25 Head 10. | (d) Layer 35 Head 10. |

Figure 8: Attention scores of Qwen3-30B-A3B. Figures (a) and (b) depict the attention score maps of the original model, where the first token clearly functions as an AS, consistently attracting the majority of attention. Figures (c) and (d) illustrate the attention scores following SE pruning, where the AS completely disappears.

which a large fraction of attention is drawn to only a few sink tokens (typically the first token). Although ASs often emerge at semantically insignificant tokens (Gu et al., 2024; Guo et al., 2024), the mechanism itself is critical for model performance. In efficient LLM techniques such as sparse attention and KV cache compression (Xiao et al., 2023b; Su & Yuan, 2025; Su et al., 2025b; Xiong et al., 2025), maintaining ASs is essential for preventing undesirable distributional shifts of attention scores.

To validate this insight and quantitatively assess the impact of SEs compression, we introduce **Attention Sinks Decay Rate**, denoted as $D_{\text{sink}}$. It is defined as the average decay rate of ASs across all heads:

$$D_{\text{sink}} = 1 - \frac{1}{H} \sum_{h=1}^{H} \frac{\sum_{i \in S} p_i^{t\prime}}{\sum_{i \in S} p_i^t} \quad (7)$$

where $H$ is the total number of heads, $p_i^t$ represents the attention score between the Query token $t$ and the Key token $i$ before SEs pruning,

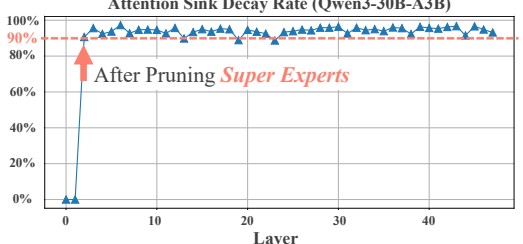

Figure 9: $D_{\text{sink}}$ of Qwen3-30B-A3B across layers.

$p_i^{t\prime}$ denotes the attention score after SEs pruning, and $S$ refers to the set of sink tokens. We evaluate $D_{\text{sink}}$ on Qwen3-30B-A3B with the C4 dataset, identifying the first token as the attention sink token. As shown in Figure 9, after SE pruning, the $D_{\text{sink}}$ remains consistently high, at approximately or even exceeding 90%, demonstrating a substantial disruptive effect on ASs. Figure 8 visualizes the attention scores for several heads before and after pruning SEs, highlighting the complete disappearance of ASs following SE pruning. Notably, ASs introduce implicit attention biases (Sun et al., 2024; An et al., 2025) that persist across all subsequent tokens and may encode global or other critical information (Darcet et al., 2023). Consequently, the impact of SEs compression on attention computation remains both continuous and significant.

## 6 RELATED WORK ON EXPERT-LEVEL COMPRESSION

M-SMoE (Li et al., 2023) performs expert merging by using activation frequencies to consolidate less significant experts, while also applying low-rank techniques to the merged experts to achieve further compression. NAEE (Lu et al., 2024) introduces plug-and-play pruning and skipping methods that leverage reconstruction loss to selectively compress less critical experts. MC (Huang et al., 2025) harnesses the significance of both experts and tokens to perform mixed-precision quantization and dynamic expert pruning, achieving extreme compression. MC-Suite (Jaiswal et al., 2025) reviews various empirical criteria for identifying critical experts, considering four dimensions: weight, expert behavior, intermediate activations, and gradient behavior. Besides pruning-based methods, there are also a few works that specifically study quantization in MoE LLMs (Duanmu et al., 2025; Zheng et al., 2025; Hu et al., 2025). While these methods examine expert importance from various perspectives to optimize expert compression, they lack a deeper exploration and understanding of the

mechanistic importance of specific experts. This study constitutes the first systematic characterization of SEs, analyzing their properties, functional impact on attention mechanisms, and contribution to systematic outliers, thereby filling a critical gap in current understanding of MoE LLMs.

## 7    CONCLUSION AND FUTURE WORK

In this work, we present the first systematic identification and comprehensive characterization of a distinct and exceptionally rare subset of experts, termed Super Experts. We thoroughly examine their distributions, intrinsic properties, and critical functional roles in driving systematic outliers. While these findings provide essential insights into the internal mechanisms of MoE LLMs, several important research directions remain open for further exploration. Specifically, future investigations will explore leveraging SEs for improved post-training compression and studying their formation during training dynamics, with the objective of mitigating extreme imbalances among experts.

## 8    ETHICS STATEMENT

This research adheres to established ethical standards in the field. All data used in experiments were obtained from publicly available sources or with appropriate permissions, and no sensitive or personally identifiable information was utilized. LLMs were employed exclusively as linguistic aids for text refinement, including grammar and stylistic improvements, and did not contribute to the design, execution, analysis, or conclusions of the study. The authors have taken care to ensure that the research findings are accurate, unbiased, and presented responsibly, with consideration for potential societal impacts.

## 9    REPRODUCIBILITY STATEMENT

All models, datasets, experimental setups, and hyperparameters used in this work are thoroughly documented. Key code components are provided in the supplementary materials, and the full algorithmic procedures are detailed in the appendix. Together, these details provide sufficient information for other researchers to independently verify and reproduce the results reported in this work.

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

## A   STATEMENT ON THE USE OF LARGE LANGUAGE MODELS

In the preparation of this paper, LLMs were employed solely as linguistic aids to enhance clarity, correctness, and readability, including grammar refinement and stylistic improvement.

## B   RELATED WORK ON SYSTEMATIC OUTLIERS IN TRANSFORMERS

Previous studies (Su & Yuan, 2025; An et al., 2025) have shown that Transformer-based dense LLMs exhibit systematic outliers. These outliers appear in multiple forms, including weight outliers (also referred to as super weights (Yu et al., 2024)), activation outliers (encompassing both activation spikes and MAs (Yang et al., 2025b; Sun et al., 2024)), and attention outliers (commonly known as attention sinks (ASs) (Xiao et al., 2023b)). This phenomenon is not confined to LLMs but is also observed in other Transformer-based architectures, including BERT (Devlin et al., 2019; Kovaleva et al., 2019; Clark et al., 2019b), Vision Transformer (ViT) (Dosovitskiy et al., 2020; Bondarenko et al., 2023; Sun et al., 2024).

Quantizable Transformers (Bondarenko et al., 2023), as a pioneering study, identified the bottleneck in activation quantization of Transformers caused by extreme outliers and revealed the intrinsic relationship between attention focus patterns and these outliers. The study further showed that attention focus emerge as attention heads attempt to perform a "no-op" or a partial update of the residual. In this process, strong activation outliers arise due to the limitations of the softmax function, which cannot produce exact zeros or ones. Consequently, Transformers learn a workaround in which attention disproportionately concentrates on a small set of fixed tokens, whose corresponding Value States typically have small norms. As a result, the attention output remains small, effectively endowing the model with a no-op capability.

Building on this insight, recent research has shown that enhancing softmax attention can substantially mitigate or eliminate systematic outliers during pretraining, thereby enabling more accurate low-precision quantization. Quantizable Transformers (Bondarenko et al., 2023) demonstrate that pretraining with clipped softmax and gated attention produces significantly smaller outliers while preserving, and in some cases even enhancing, floating-point task performance. Qwen Team (Qiu et al., 2025) finds that applying a head-specific sigmoid gate after the Scaled Dot-Product Attention (SDPA) consistently improves performance and eliminates systematic outliers. Softpick (Zuhri et al., 2025) introduces a rectified drop-in replacement for softmax in Transformer attention that relaxes the sum-to-one constraint, effectively eliminates attention sink and massive activations, and holds strong promise for advancing quantization, low-precision training, sparsity optimization, pruning, and interpretability. By revealing SEs as the root source of systematic outliers, this work provides the first comprehensive characterization of such phenomena in MoE LLMs and establishes a foundation for future advances in outlier mitigation.

## C   FURTHER ANALYSIS OF OUTLIER EXPERTS IN FINAL LAYERS

Some experts in the final layers also exhibit extreme activation outliers in the output of the down_proj, apart from the SEs in the shallower layers. We refer to these experts as outlier experts. Based on our extensive additional experiments and findings, we do not consider outlier experts to have the same mechanistic significance as SEs:

*(i)* We performed PPL evaluations after pruning outlier experts, and as shown in Table 6, they do not significantly affect the model's performance in the same way as the SEs.

*(ii)* In both Qwen3-30B-A3B and DeepSeek-R1, pruning outlier experts does not result in repetitive outputs on reasoning benchmarks such as Math-500 (Lightman et al., 2023), whereas pruning SEs does, as shown in Tables 11 and 12. The pruned SEs and outlier experts is shown in Table 7.

*(iii)* We observed on Qwen3-30B-A3B that the distribution of outlier experts varies with the input dataset, while the distribution of SEs remains quite stable, as illustrated in Figure 14 and 15.

*(iv)* The router scores in the final layer are relatively evenly distributed across all experts for both sink and non-sink tokens, as shown in Figure 10. In contrast, sink-token router logits are strongly

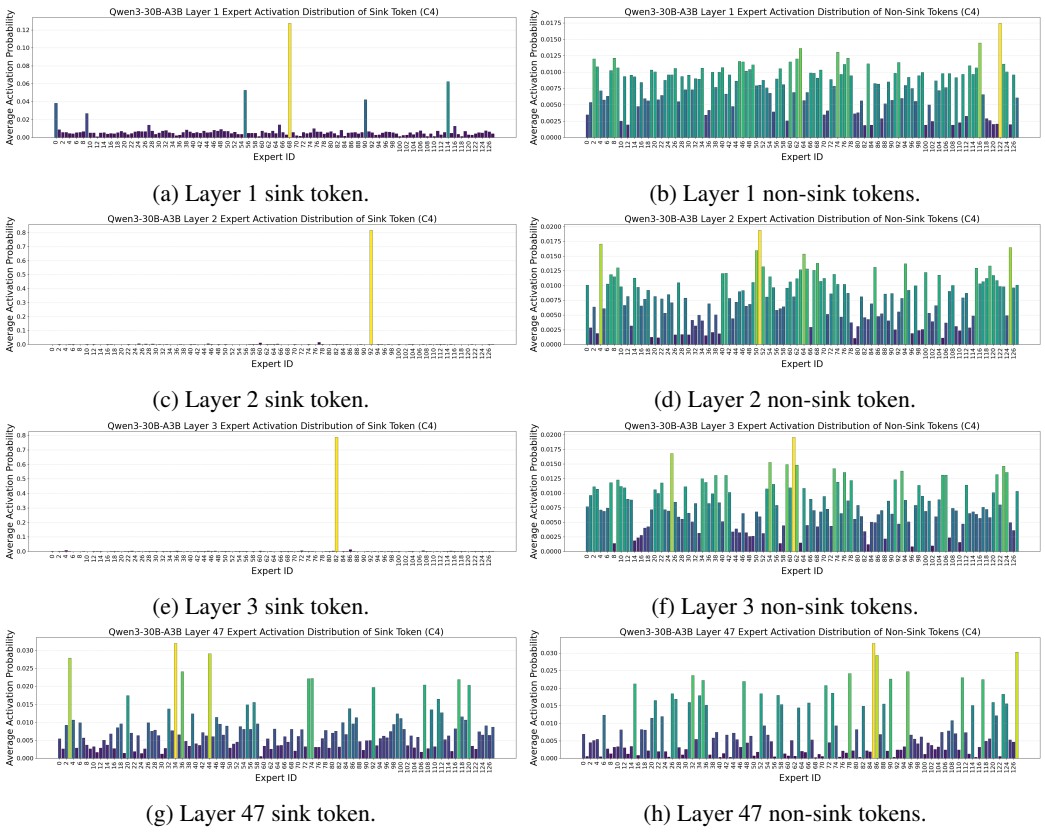

Figure 10: Expert router score distributions for sink and non-sink tokens in Qwen3-30B-A3B, based on calibration using the C4 dataset.

skewed toward SEs, indicating that the final-layer outlier experts are fundamentally different from SEs.

We infer that since MAs occur in the shallower layers, these outlier experts are not involved in the formation of MAs. Therefore, these experts do not operate under the same mechanism as SEs and do not hold the same level of significance.

# D    DISTRIBUTION OF SUPER EXPERTS ACROSS VARIOUS DATA DOMAINS

In addition to analyzing the distribution of SEs across different models based on the C4 dataset (Raffel et al., 2020), we also examine their distribution patterns across various input data domains. We assess the impact of diverse language inputs on SEs using the WikiText-2 (Merity et al., 2016) and C-Eval (Huang et al., 2023) datasets. Furthermore, we investigate the influence of data from the mathematics and coding domains using the GSM8K (Hendrycks et al., 2021) and HumanEval (Chen et al., 2021) datasets. As shown in Figures 14, 15, 16, 17, 18, and 19, the distribution of SEs remains highly stable, regardless of variations in the input data domain.

# E    ADDITIONAL RESULTS OF REASONING MODELS AFTER SUPER EXPERTS PRUNING

After pruning SEs, we consistently observed repetitive output and a loss of reasoning ability in both Qwen3-30B-A3B and DeepSeek-R1. The pruned SEs are shown in Table 7, and additional examples from the Math-500 (Lightman et al., 2023) benchmark are presented in Tables 11 and 12.

Table 6: Comparison of expert pruning, with PPL evaluated using the WikiText-2 dataset.

| Model | Prune Experts | PPL | Super Experts |
|-------|---------------|-----|---------------|
| Qwen3-30B-A3B | Original Model | 8.70 | - |
| | Layer 1 Expert 68, Layer 2 Expert 92, Layer 3 Expert 82 | 59.86 | Yes |
| | Layer 47 Expert 8, Layer 47 Expert 48, Layer 47 Expert 100 | 8.71 | No |
| DeepSeek-V2-Lite | Original Model | 6.31 | - |
| | Layer 3 Expert 54, Layer 4 Expert 38 | 10.75 | Yes |
| | Layer 25 Expert 11, Layer 25 Expert 39 | 6.32 | No |

Table 7: Super Experts and Outlier Experts in Qwen3-30B-A3B and DeepSeek-R1 models.

| Model | Super Experts | Outlier Experts |
|-------|---------------|-----------------|
| Qwen3-30B-A3B | Layer 1 Expert 68, Layer 2 Expert 92 
 Layer 3 Expert 82 | Layer 1 Expert 8, Layer 47 Expert 48 
 Layer 47 Expert 100 |
| DeepSeek-R1 | Layer 8 Expert 24, Layer 8 Shared_expert 
 Layer 12 Expert 190, Layer 13 Expert 64 
 Layer 14 Expert 202, Layer 14 Shared_expert 
 Layer 22 Shared_expert, Layer 33 Expert 64 
 Layer 33 Shared_expert, Layer 35 Shared_expert | Layer 60 Expert 81, Layer 60 Expert 92 
 Layer 60 Expert 231, Layer 60 Shared_expert 
 Layer 60 Expert 121, Layer 60 Expert 0 
 Layer 60 Expert 60, Layer 60 Expert 237 
 Layer 60 Expert 53, Layer 60 Expert 117 |

## F    ADDITIONAL RESULTS ON ROUTER SCORE DISTRIBUTIONS

The router score distributions calibrated on the C4 dataset are shown in Figure 20. Similarly, the router score distributions calibrated on the Wikitext-2 dataset are presented in Figure 21. Notably, in both datasets, SEs are consistently and strongly activated on the sink tokens.

## G    MORE ANALYSIS ON SUPER EXPERTS MECHANISM

Figure 22 provides a comprehensive illustration of the systematic outlier mechanism in Qwen3-30B-A3B, showing the stepwise formation process across three layers. Furthermore, the mapping from massive activations to attention sinks remains consistent even after multiple transformations, such as layer normalization (LN) and QKV projections. Drawing on prior research (Sun et al., 2024; Su & Yuan, 2025), we offer a more detailed analysis to elucidate this mechanism. Specifically, this process is governed by two key mechanisms:

**QKV suppression.** The presence of massive activations with large magnitudes results in substantially smaller normalized values for the corresponding tokens after LN, as dictated by the RMSNorm process. This reduction in norm is preserved throughout the QKV states. As shown in Figures 11, the Queries, Keys, and Values of sink tokens consistently exhibit significantly smaller norms compared to non-sink tokens.

**High cosine similarity of QK.** Despite the reduced norms of Queries and Keys, the cosine similarity between the Queries of non-sink tokens and the Keys of sink tokens remains high (Gu et al., 2024), leading to disproportionately large attention scores, as illustrated in Figure 11a.

These intermediate mechanisms ensure that massive activations remain aligned with attention sink tokens, thereby establishing their participation in systematic outliers.

## H    WEIGHT-LEVEL ANALYSES OF SUPER EXPERTS

While the preceding discussion primarily addressed the role of experts in expert-level compression, this section provides a more granular, weight-level analysis of SEs. Conducting analyses at the weight level offers several key advantages: *(i)* it clarifies the specific sources of SEs' importance, *(ii)* it facilitates the investigation of analogous patterns in dense models, and *(iii)* it enables the

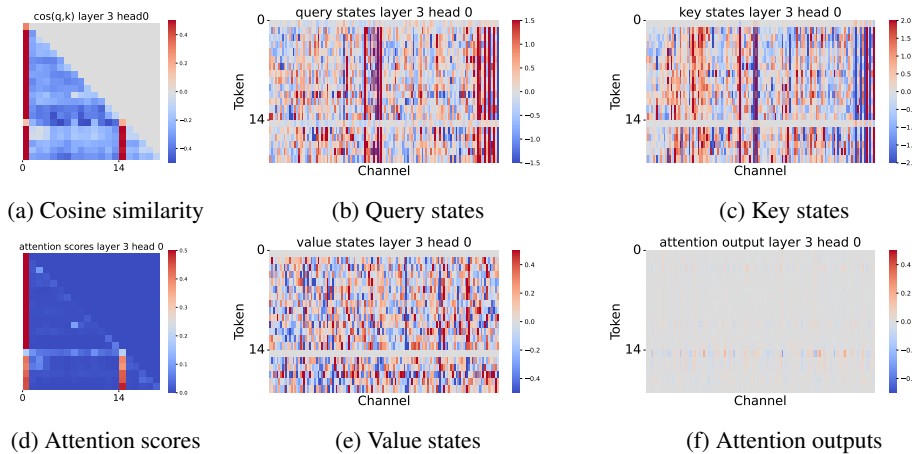

(a) Cosine similarity      (b) Query states      (c) Key states

(d) Attention scores      (e) Value states      (f) Attention outputs

Figure 11: (11b), (11c), and (11e) illustrate QKV suppression. (11a) highlights the high cosine similarity of QK. (11f) visualizes the attention output. Visualizations use the following input from MMLU (Hendrycks et al., 2021), evaluated on Llama-2-7B: "The following are multiple-choice questions (with answers) about machine learning. \n\n..."

Table 8: Super weights of several models.

| Models | Total Layers | Emergence Layer | Super Experts | Shape of down_proj | Super Weights |
|---|---|---|---|---|---|
| LLaVA-V1.5-7B | 0-31 | 1 | - | (4096, 11008) | (1415, 7890), (2533, 7890) |
| Llama-3.2-1B | 0-15 | 1 | - | (2048,8192) | (400, 1417), (698, 1417), (2029, 1417), (1159, 1417) |
| Qwen3-30B-A3B | 0-47 | 1
2
3 | 68
92
82 | (2048,768) | (940, 711)
(940, 59)
(940, 423) |

application of interpretability tools, such as sparse autoencoders (Bloom et al., 2024; Gao et al., 2024a), to examine individual weights or neurons.

Specifically, we detect extreme activation outlier channels in the down_proj inputs and outputs and map them to their corresponding weights, following the methodology used to identify Super Weights (SWs) in (Yu et al., 2024). For convenience, we refer to these weights as SWs. The SWs utilized in our experiments are listed Table 8.

These SWs are subsequently pruned during inference, and experiments are conducted on LLaVA-V1.5-7B (Liu et al., 2023), Llama3.2-1B (Touvron et al., 2023), and Qwen3-30B-A3B, spanning dense LLMs, vision-language models (VLMs), and MoE LLMs. Tables 13, 14, 15, and 16 show that removing the SWs consistently leads to repetitive and uninformative outputs across all models, demonstrating that these weights are critical contributors to SE importance. This outcome is expected, as it is consistent with our analysis of SEs' role in driving systematic outliers in Transformers. Interestingly, in dense LLMs, the single FFN layer exhibiting massive activations can be seen as playing a role analogous to that of an SE.

Why does the model exhibit a substantial performance gap before and after SWs pruning? We then investigate the underlying causes of this discrepancy. As model neurons are often polysemantic (Gao et al., 2024a), directly analyzing the semantics of SWs is challenging due to their intrinsic polysemanticity. To address this, we employ Sparse Autoencoders (SAEs) (Bloom et al., 2024; Gao et al., 2024a), an unsupervised method designed to extract interpretable features from LLMs by reconstructing activations through a sparse bottleneck layer. Using SAEs, we can decompose the semantics of polysemantic neurons into more discrete, monosemantic features and conduct semantic analyses on the features most strongly correlated with SWs. We then train our SAE using activations from the Layer 1 FFN outputs of Llama-3.2-1B with the C4 dataset (Raffel et al., 2020). The loss function consists of two components: a reconstruction loss and a sparsity penalty loss (Bloom et al.,

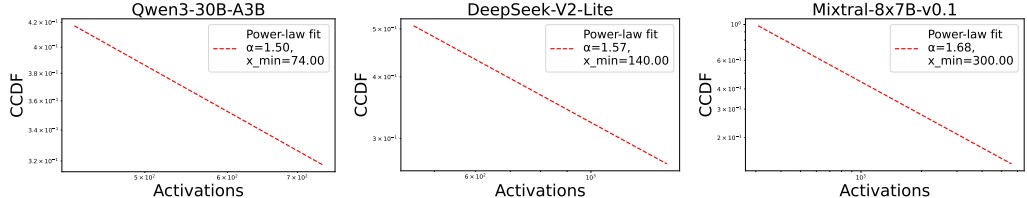

Figure 12: CCDF analysis of the distribution of $\mathcal{A}$ for Qwen3-30B-A3B, Mixtral-8×7B, and DeepSeek-V2-Lite on the C4 dataset. Across all three models, the fitted CCDF tails exhibit a clear power-law form, indicating that only a small subset of experts produce exceptionally large activations. For each model, the estimated tail exponent $\alpha$ is below 2, placing the distribution within the heavy-tailed regime. This establishes the presence of extreme yet statistically stable structured activation outliers in $\mathcal{A}$.

2024). The reconstruction loss is defined as

$$mse\_loss = \frac{1}{N \cdot D} \sum_{i=1}^{N} \sum_{j=1}^{D} \left( \text{sae\_out}_{i,j} - \text{sae\_in}_{i,j} \right)^2, \tag{8}$$

where $N$ denotes the batch size and $D$ the dimensionality of the activations. The sparsity penalty loss is given by

$$l1\_loss = \lambda \cdot \frac{1}{N} \sum_{i=1}^{N} \left( \sum_{k=1}^{K} \text{feature\_acts}_{i,k} \cdot \|\text{W\_dec}_k\|_2 \right)^{1/p}, \tag{9}$$

where $\lambda$ is a hyperparameter controlling sparsity, $K$ is the number of features in the bottleneck layer, feature_acts$_{i,k}$ denotes the activation of feature $k$ for sample $i$, W_dec$_k$ is the corresponding decoder weight vector, and $p$ specifies the norm used for aggregation.

After achieving satisfactory performance with the SAE, we apply the TopK algorithm to the decoder weight matrix to extract the ten features most strongly associated with the SWs neurons. Each feature is interpreted via a forward pass to infer its semantic meaning (McDougall, 2024). As illustrated in Figure 23, the top features associated with the SWs neuron consistently exhibit pronounced activation at the `end_of_text` token, a pattern that is rarely observed in other neurons. This finding provides a plausible explanation for the behavior of LLMs, which repeatedly generate text until reaching the maximum output length following the pruning of SEs or SWs. When SEs are pruned, MoE LLMs lose the ability to recognize sentence boundaries and generate text continuously until reaching the output length limit. This indicates that SEs play a critical role in regulating sentence length and termination. We further hypothesize that SEs may contribute to additional model capabilities, which we will explore in future research.

## I  ANALYSIS OF THRESHOLD-BASED SUPER EXPERTS IDENTIFICATION

In this section, we demonstrate that SEs reflect an intrinsic property of MoE LLMs, with threshold-based identification serving as a practical and principled detection method. We first provide a justification based on the heavy-tailed distribution of $\mathcal{A} = \{a_{l,e}\}$ (as discussed in Section 3.2.1), and then analyze the robustness of the threshold-based SE identification.

The identification of SEs is motivated by the heavy-tailed nature of extreme activations. While most experts produce modest responses, a small subset consistently generates extreme values (as illustrated in the heatmaps in Figure 5 and Appendix D), forming a distinct long tail that is naturally present in MoE LLMs. To mathematically validate this observation, we analyze the tail of the activation distribution using standard heavy-tail analysis techniques, namely the complementary cumulative distribution function (CCDF) and power-law fitting. The CCDF of a random variable $X$ is defined as

$$\text{CCDF}(x) = P(X > x), \tag{10}$$

Table 9: Sensitivity analysis of SE identification across a range of threshold settings for multiple MoE LLMs and input domains.

| Models | Number of Identified Experts | | | | | | | | | |
|---|---|---|---|---|---|---|---|---|---|---|
| | C4 | $P_{95}$ | $P_{99}$ | $P_{99.5}$ | $P_{99.9}$ | WikiText-2 | $P_{95}$ | $P_{99}$ | $P_{99.5}$ | $P_{99.9}$ |
| Qwen3-30B-A3B | $0.1 * a_{\max}$ | 3 | 3 | 3 | 3 | $0.1 * a_{\max}$ | 3 | 3 | 3 | 3 |
| | $0.09 * a_{\max}$ | 3 | 3 | 3 | 3 | $0.09 * a_{\max}$ | 3 | 3 | 3 | 3 |
| | $0.08 * a_{\max}$ | 3 | 3 | 3 | 3 | $0.08 * a_{\max}$ | 3 | 3 | 3 | 3 |
| | $0.07 * a_{\max}$ | 3 | 3 | 3 | 3 | $0.07 * a_{\max}$ | 3 | 3 | 3 | 3 |
| | C4 | $P_{95}$ | $P_{99}$ | $P_{99.5}$ | $P_{99.9}$ | WikiText-2 | $P_{95}$ | $P_{99}$ | $P_{99.5}$ | $P_{99.9}$ |
| DeepSeek-V2-Lite | $0.1 * a_{\max}$ | 2 | 2 | 2 | 2 | $0.1 * a_{\max}$ | 2 | 2 | 2 | 2 |
| | $0.09 * a_{\max}$ | 2 | 2 | 2 | 2 | $0.09 * a_{\max}$ | 2 | 2 | 2 | 2 |
| | $0.08 * a_{\max}$ | 2 | 2 | 2 | 2 | $0.08 * a_{\max}$ | 2 | 2 | 2 | 2 |
| | $0.07 * a_{\max}$ | 2 | 2 | 2 | 2 | $0.07 * a_{\max}$ | 2 | 2 | 2 | 2 |
| | C4 | $P_{95}$ | $P_{99}$ | $P_{99.5}$ | $P_{99.9}$ | WikiText-2 | $P_{95}$ | $P_{99}$ | $P_{99.5}$ | $P_{99.9}$ |
| Mixtral-8x7B | $0.1 * a_{\max}$ | 1 | 1 | 1 | 1 | $0.1 * a_{\max}$ | 1 | 1 | 1 | 1 |
| | $0.09 * a_{\max}$ | 1 | 1 | 1 | 1 | $0.09 * a_{\max}$ | 1 | 1 | 1 | 1 |
| | $0.08 * a_{\max}$ | 1 | 1 | 1 | 1 | $0.08 * a_{\max}$ | 1 | 1 | 1 | 1 |
| | $0.07 * a_{\max}$ | 1 | 1 | 1 | 1 | $0.07 * a_{\max}$ | 1 | 1 | 1 | 1 |

representing the probability that $X$ exceeds a given value $x$. Plotting the CCDF on a log-log scale provides an initial visual assessment of the tail behavior. To quantitatively characterize the tail, we fit it to a power-law model

$$P(X > x) \propto x^{-\alpha}, \quad x \geq x_{\min}, \tag{11}$$

where $\alpha$ is the tail exponent and $x_{\min}$ denotes the minimum value above which the power-law behavior holds. The exponent $\alpha$ is estimated using maximum likelihood estimation (MLE). As shown in Figure 12, the CCDF tail is consistent with a power-law model. This long-tail behavior indicates that only a few experts dominate the extreme activations, naturally separating them from the majority and confirming that extreme activations follow an inherent heavy-tailed distribution rather than arising from random noise. Threshold-based methods thus provide a practical and principled approach for identifying this small, statistically significant subset of experts.

To further assess the robustness of SE identification, we perform a sensitivity analysis over a reasonable range of thresholds across different MoE LLMs and input domains. As shown in Table 9, SE identification remains consistent, confirming that extreme activation outliers reflect an intrinsic property of MoE LLMs rather than an artifact of any specific threshold choice. Together, these analyses demonstrate that the threshold-based criterion is both robust and scientifically justified, and that SEs constitute an inherent feature of MoE LLMs.

## J CROSS-DOMAIN ANALYSIS OF SUPER EXPERTS AND ATTENTION SINK TOKENS

In this section, we conduct a cross-domain analysis of SEs and attention sink tokens using inputs drawn from diverse domains, including out-of-distribution (OoD) inputs that differ substantially from the training datasets. Each domain includes visualizations of SE heatmaps, attention sink patterns, and router score distributions for both sink and non-sink tokens.

We use C4, C-Eval, GSM8K, and HumanEval (Raffel et al., 2020; Huang et al., 2023; Hendrycks et al., 2021; Chen et al., 2021), covering a range of domains including English, Chinese, mathematics, and code. For the OoD datasets, we adopt Pile-of-Law and MedDialog (Henderson* et al., 2022; Chen et al., 2020), which are legal and medical datasets and differ significantly from the pretraining corpus of the tested MoE LLMs in content and domain.

As shown in Figures 24, 25, 26, 27, 28 and 29, the distribution of SEs remains highly consistent across these varied domains and OoD datasets. Moreover, the sink tokens consistently appear as the first token. Their router scores on SEs are markedly larger, whereas the router scores of non-sink tokens remain relatively uniform across experts. These observations reinforce our claim that

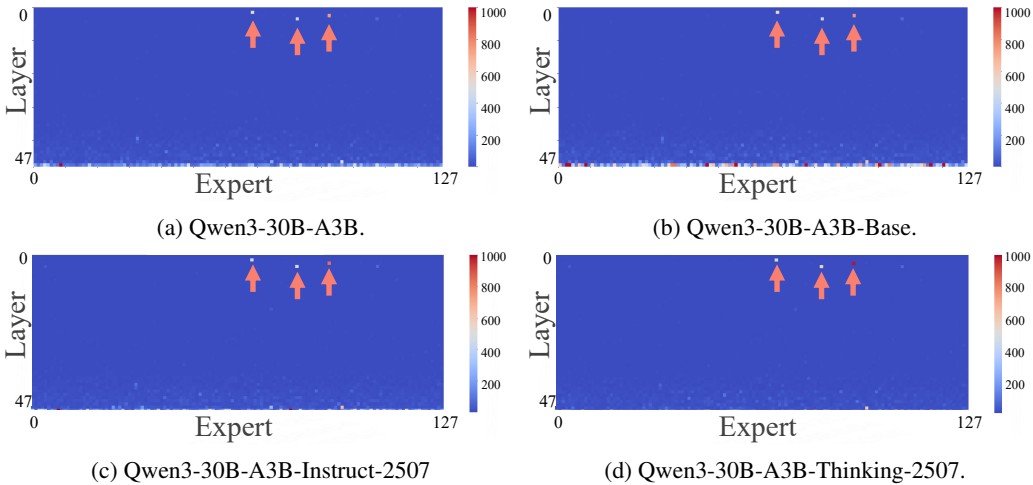

Figure 13: Heatmap visualizations of the maximum output magnitudes from the `down_proj` for each expert across layers based on C4 dataset. SEs are highlighted with arrows.

Table 10: Super Experts in Qwen3-30B-A3B.

| Models | Super Experts |
|---|---|
| Qwen3-30B-A3B-Base Qwen3-30B-A3B Qwen3-30B-A3B-Instruct-2507 Qwen3-30B-A3B-Thinking-2507 | Layer 1 Expert 68, Layer2 Expert 92, Layer3 Expert 82 |

SEs reflect an intrinsic property of MoE LLMs, rather than arising from over-exposure to domain-specific tokens, and are not affected by OoD inputs.

## K    DISTRIBUTION OF SUPER EXPERTS ACROSS TRAINING STAGES

To study the distribution of SEs across training stages, we identified SEs in four officially open-sourced Qwen3-30B-A3B models at different training stages: Qwen3-30B-A3B-Base, Qwen3-30B-A3B, Qwen3-30B-A3B-Instruct-2507, and Qwen3-30B-A3B-Thinking-2507. The indices of the SEs are shown in Table 10, and the SE heatmaps are presented in Figure 13, demonstrating fully consistent sets across all models and training stages. These results support the conclusion that SEs are persistent features of MoE LLMs.

## L   ALGORITHM FOR PROFILING SUPER EXPERTS

The detailed procedure for profiling Super Experts is illustrated in Algorithm 1.

---

**Algorithm 1** Calibration-based Super Experts Profiling

---

 1: **Input:** Model with $E$ experts per layer, calibration dataset $\mathcal{D}$
 2: **Output:** Set of Super Experts $S$

 3: **Stage 1: Calibration of MA-formation Layers**
 4: $L \leftarrow \emptyset$
 5: **for** each batch $x \in \mathcal{D}$ **do**
 6:     **for** each layer $l$ in the model **do**
 7:         Compute hidden activations $H^l(x)$
 8:         **if** MA pattern detected in $H^l(x)$ **then**
 9:             $L \leftarrow L \cup \{l\}$
10:         **end if**
11:     **end for**
12: **end for**

13: **Stage 2: Identification of Super-Experts**
14: $\mathcal{A} \leftarrow \emptyset$
15: **for** each batch $x \in \mathcal{D}$ **do**
16:     **for** each layer $l \in L$ **do**
17:         **for** each expert $e$ in layer $l$ **do**
18:             Compute output $h_{l,e}(x)$ before `down_proj`
19:             $a_{l,e} \leftarrow \max_{x \in \mathcal{D}} |h_{l,e}(x) \cdot W^{l,e}_{\text{down\_proj}}|$
20:             $\mathcal{A} \leftarrow \mathcal{A} \cup \{a_{l,e}\}$
21:         **end for**
22:     **end for**
23: **end for**
24: $P_{99.5} \leftarrow \text{Percentile}_{99.5}(\mathcal{A})$
25: $a_{\max} \leftarrow \max(\mathcal{A})$
26: $S \leftarrow \emptyset$
27: **for** each $(l, e)$ with $a_{l,e} \in \mathcal{A}$ **do**
28:     **if** $a_{l,e} > P_{99.5}$ **and** $a_{l,e} > \frac{1}{10} a_{\max}$ **then**
29:         $S \leftarrow S \cup \{(l, e)\}$
30:     **end if**
31: **end for**
32: **return** $S$

---

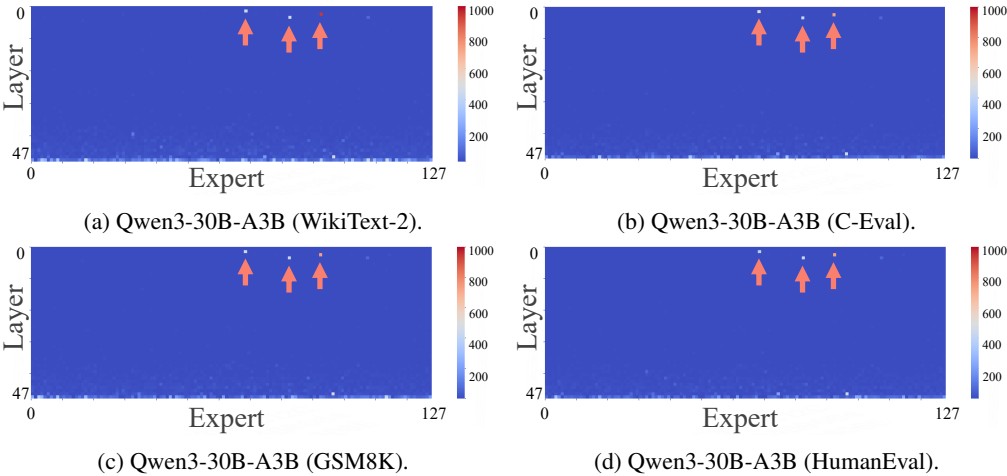

Figure 14: Heatmap visualizations of the maximum output magnitudes from the down_proj for each expert in Qwen3-30B-A3B across multiple datasets. SEs are highlighted with arrows.

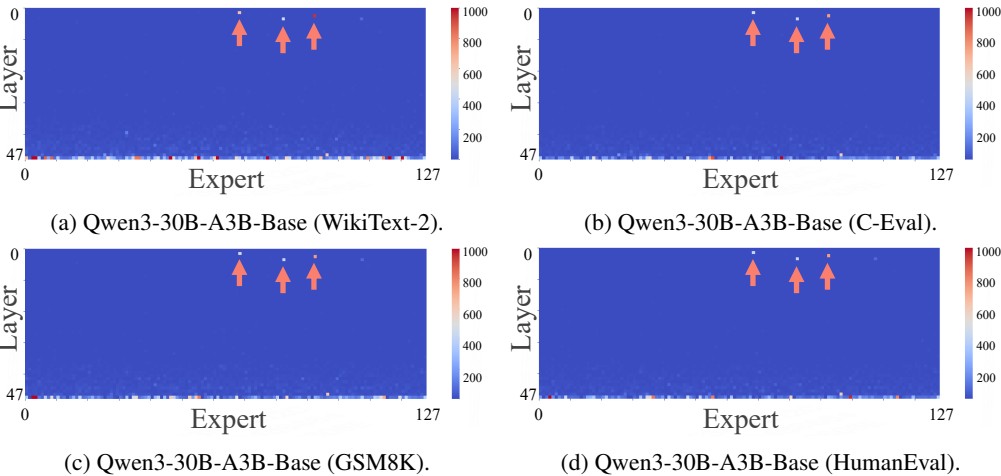

Figure 15: Heatmap visualizations of the maximum output magnitudes from the down_proj for each expert in Qwen3-30B-A3B-Base across multiple datasets. SEs are highlighted with arrows.

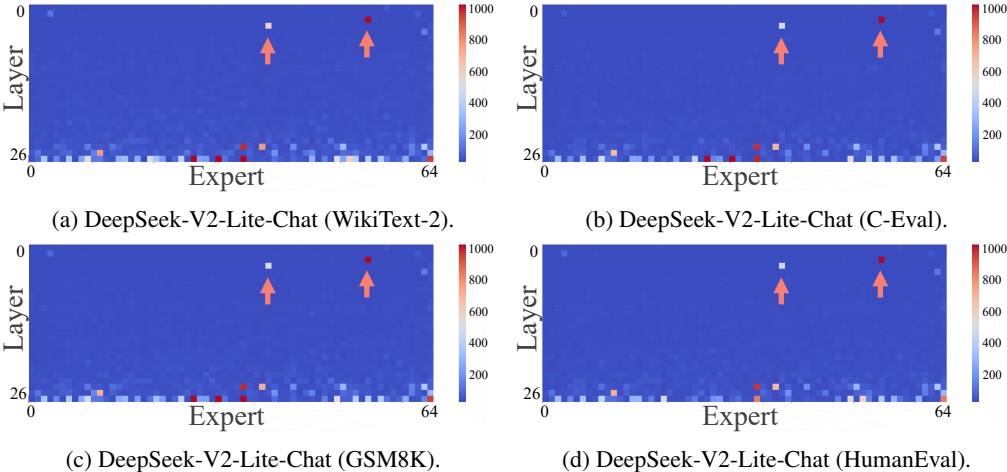

Figure 16: Heatmap visualizations of the maximum output magnitudes from the `down_proj` for each expert in DeepSeek-V2-Lite-Chat across multiple datasets. SEs are highlighted with arrows.

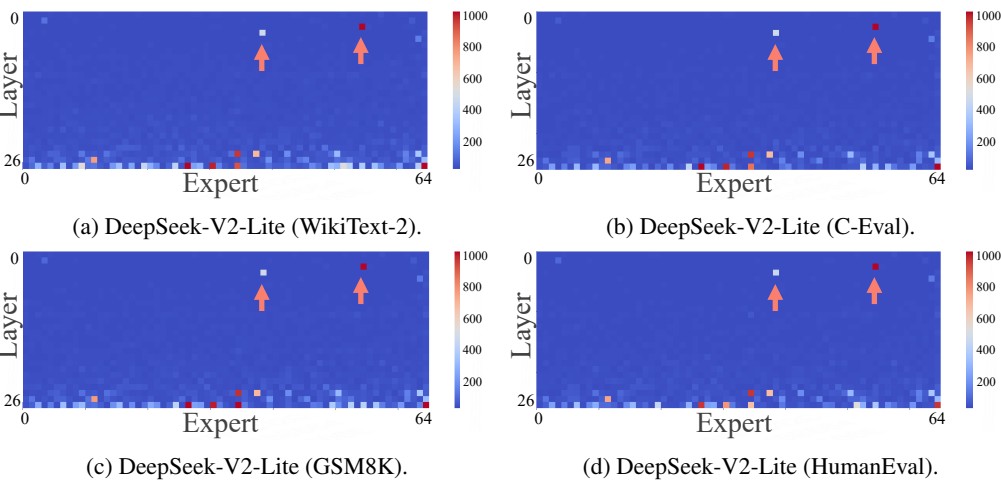

Figure 17: Heatmap visualizations of the maximum output magnitudes from the `down_proj` for each expert in DeepSeek-V2-Lite across multiple datasets. SEs are highlighted with arrows.

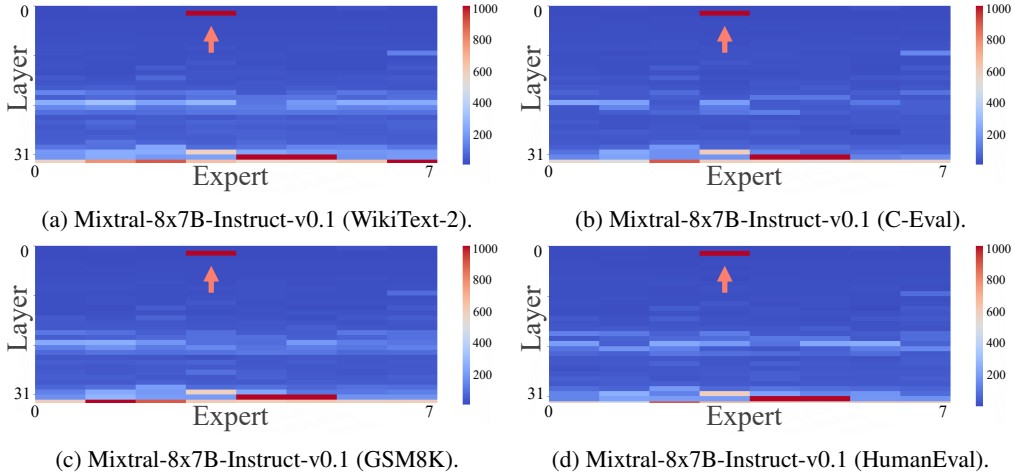

Figure 18: Heatmap visualizations of the maximum output magnitudes from the `down_proj` for each expert in Mixtral-8x7B-Instruct-v0.1 across multiple datasets. SEs are highlighted with arrows.

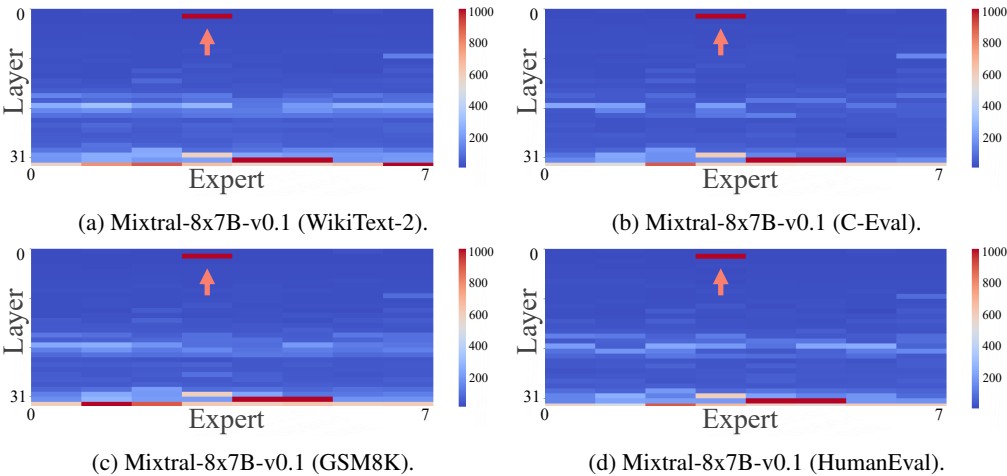

Figure 19: Heatmap visualizations of the maximum output magnitudes from the `down_proj` for each expert in Mixtral-8x7B-v0.1 across multiple datasets. SEs are highlighted with arrows.

Table 11: Responses of DeepSeek-R1 in the MATH-500 benchmark.

| DeepSeek-R1 | Input | Repeating | Answer |
|---|---|---|---|
| Original Model | What is the domain of the function $f(x) = \frac{2-x}{\log(2-\log(x-2))}$, where log is the base 10 logarithm function? Express your answer in interval notation. Please reason step by step, and put your final answer within \boxed{}. | no | Okay, so I need to find the domain of the function f(x) = (2 - x) / log(2 - log(x - 2)). Hmm, let's see. The domain of a function includes all the real numbers x ... ... |
| Prune Super Experts | | yes | Alright, let's try to first. the function f(x) which given that we need to the domain of this function that's the which the of a that's the given this. that's the the the the the the the the the the the the the the the. that's the. that's the. that's the. that's the. that's the. ... ... Repeating ... ... that's the. that's the. that's the. that's the. that's the. |
| Prune Outlier Experts of Last Layer | | no | Okay, let's try to figure out the domain of this function $f(x) = \frac{2-x}{\log(2-\log(x-2))}$. Hmm, domains can be tricky because there are multiple layers ... ... |
| Random Pruning | | no | Alright, so I need to find the domain of the function f(x) = (2 - x) / log(2 - log(x - 2)). Hmm, okay. Let me start by recalling that the domain of a ... ... |

Table 12: Responses of Qwen3-30B-A3B in the MATH-500 benchmark after SE pruning.

| Qwen3-30B-A3B | Input | Repeating | Answer |
|---|---|---|---|
| Original Model | Let $p(x)$ be a polynomial of degree 5 such that $p(n) = \frac{n}{n^2-1}$ for $n = 2, 3, 4, \ldots, 7$. Find $p(8)$. Please reason step by step, and put your final answer within \boxed{}. | no | Okay, so I need to find the value of p(8) where p(x) is a degree 5 polynomial that satisfies p(n) = n / (n² - 1) for n = 2, 3, 4, ..., 7. ... ... |
| Prune Super Experts | | yes | Okay, so I need to find p(8) where p is a polynomial. The problem says that for each n, p(n) is equal to some fraction, and we need to find p(8). The question is from a competition, so it's probably from an Olympiad, and the answer is an integer. But I need to reason step by step, and put my final answer within \boxed{}. Alright, let's start by understanding the problem. It says that for each n, p(n) is equal to some fraction. But the problem is to find p(8). The question is from a competition, so it's probably from an Olympiad, and the answer is an integer. But I need to reason step by step, and put my final answer within ... ... Repeating ... ... |
| Prune Outlier Experts of Last Layer | | no | Okay, so I need to find the value of p(8) where p(x) is a degree 5 polynomial that satisfies p(n) = n / (n² - 1) for n = 2, 3, 4, 5, 6, 7. Hmm, let me think about how ... ... |
| Random Pruning | | no | Okay, so I need to find the value of p(8) where p(x) is a degree 5 polynomial that satisfies p(n) = n / (n² - 1) for n = 2, 3, 4, ..., 7. ... ... |

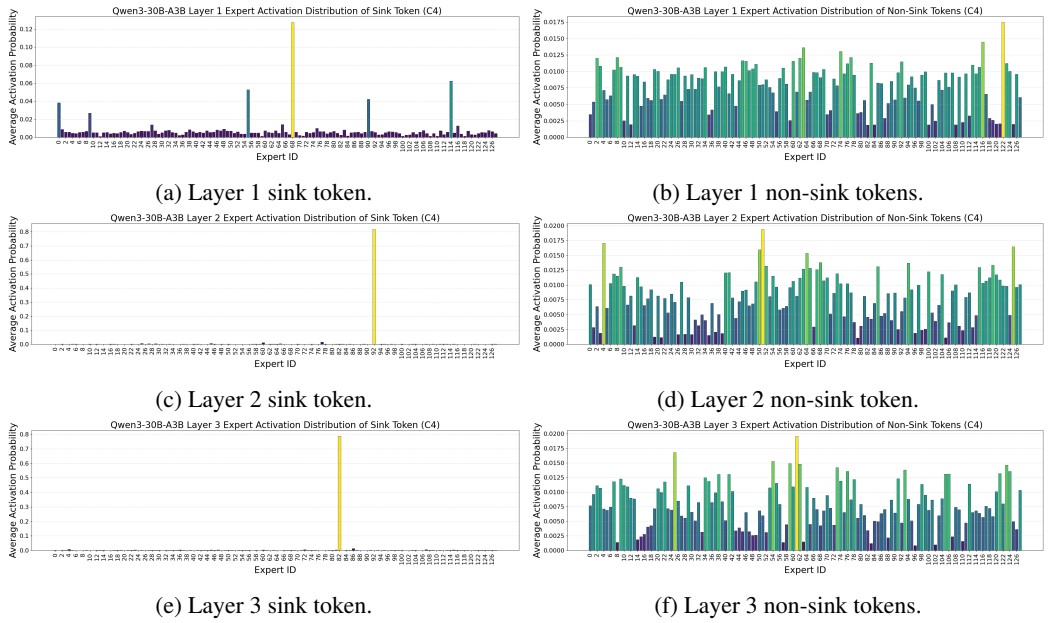

Figure 20: Expert router score distributions for sink and non-sink tokens in Qwen3-30B-A3B, based on calibration using the C4 dataset.

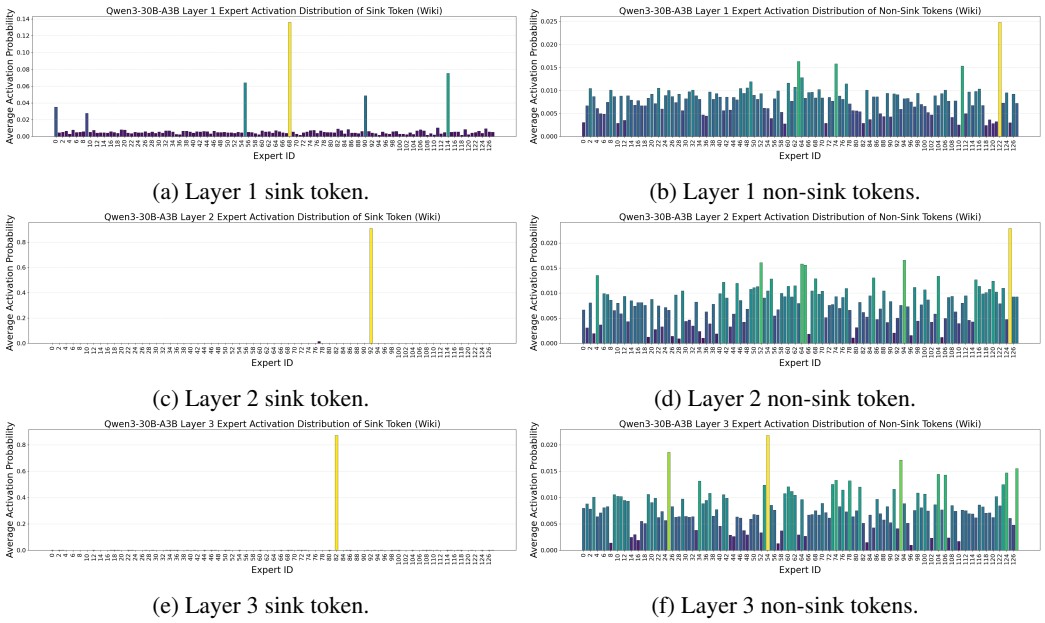

Figure 21: Expert router score distributions for sink and non-sink tokens in Qwen3-30B-A3B, based on calibration using the Wikitext-2 dataset.

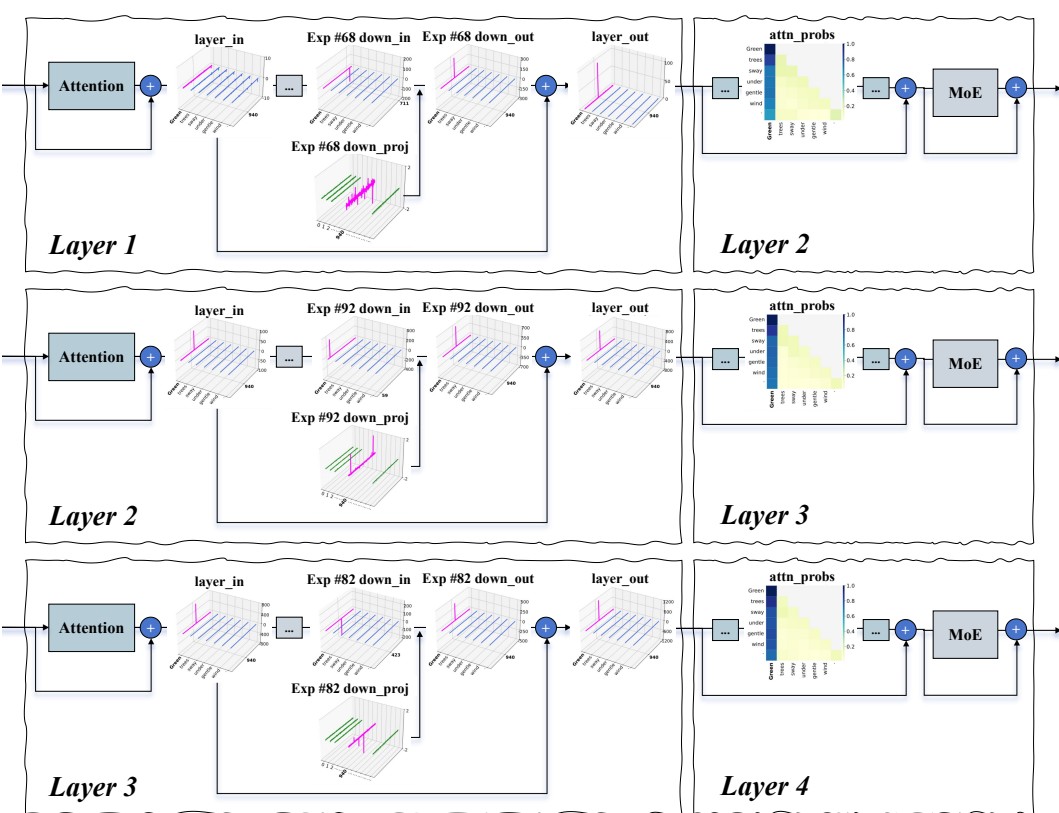

Figure 22: Systematic outlier mechanism of Qwen3-30B-A3B, using the input: "Green trees sway under gentle wind."

Table 13: Responses of LLaVA-V1.5-7B after SWs pruning.

| LLaVA-V1.5-7B | Input | Repeating | Answer |
|---|---|---|---|
| Original Model | | no | The player holding the baseball bat is now wearing the number 44. |
| Prune SWs Neurons | What number is the player holding the baseball bat now? | yes | OpOpquinquinquinOpOpOpOpOpOp ... |
| Random Pruning | | no | The player holding the baseball bat is now wearing the number 44. |

Table 14: Responses of LLaVA-V1.5-7B after SWs pruning.

| LLaVA-V1.5-7B | Input | Repeating | Answer |
|---|---|---|---|
| Original Model | | no | The traffic sign says "You Can't Stop Obama." |
| Prune SWs Neurons | What does the traffic sign say? | yes | \\\\*\\\\\\\\\\\\\\\\\\ ... |
| Random Pruning | | no | The traffic sign says "You Can't Stop Obama." |

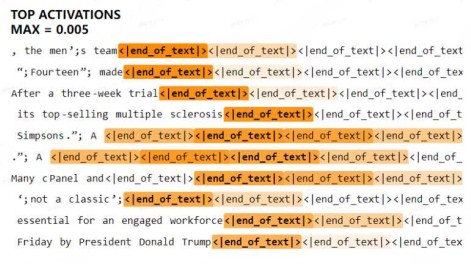

(a) The top features associated with the SWs neuron consistently exhibit pronounced activation at the `end_of_text` token.

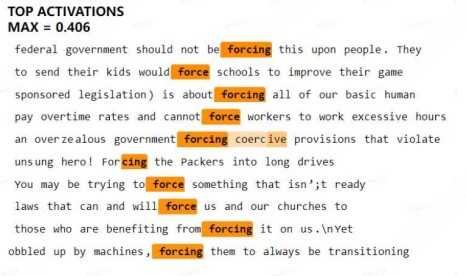

(b) The top features in other neurons.

Figure 23: Top features identified in SW neurons versus other neurons using the trained Sparse Autoencoder.

Table 15: Responses of Llama-3.2-1B after SWs pruning.

| Llama-3.2-1B | Input | Repeating | Answer |
|---|---|---|---|
| Original Model | Solve for x: 3x + 7 = 25. | no | 3x = 18. x = 6 Step 1: Write down the equation given in the problem. 3x + 7 = 25. Step 2: Subtract 7 from both sides of the equation to isolate the term with x. 3x + 7 - 7 = 25 - 7, which simplifies to 3x = 18. Step 3: Divide both sides of the equation by 3 to solve for x. 3x / 3 = 18 / 3, which simplifies to x = 6. The final answer is: 6 |
| Prune SWs Neurons | | yes | sselsselsselsselsselsselsselsselsselsey Quy Quy Quy Quy Quy Quy Quy Quy Quy Quy Quy Quy Quy Quy Quy Quy Quy Quy Quy Quy Quy Quy Quy Quy Quy Quy Quy Quy Quy Quy Quy Quy Quy Quy Quy Quy Quy Quy Quy Quy Quy Quy Quy Quy Quy Quy Quy Quy Quy Quy Quy Quy Quy Quy Quy Quy Quy Quy Quy Quy Quy Quy Quy Quy Quy Quy Quy Quy Quy Quy Quy Quy Quy Quy Quy Quy Quy Quy Quy Quy Quy Quy Quy Quy Quy Quy Quy Quy Quy Quy Quy Quy Quy Quy Quy Quy Quy Quy Quy Quy Quy Quy Quy Quy Quy Quy Quy Quy Quy Quy Quy Quy Quy Quy Quy Quy Quy Quy Quy Quy Quy Quy Quy Quy Quy Quy ... |
| Random Pruning | | no | 3x = 18. x = 6 Step 1: Write down the equation given in the problem. 3x + 7 = 25. Step 2: Subtract 7 from both sides of the equation to isolate the term with x. 3x + 7 - 7 = 25 - 7, which simplifies to 3x = 18. Step 3: Divide both sides of the equation by 3 to solve for x. 3x / 3 = 18 / 3, which simplifies to x = 6. The final answer is: 6 |

Table 16: Responses of Qwen3-30B-A3B after SWs pruning.

| Qwen3-30B-A3B | Input | Repeating | Answer |
|---|---|---|---|
| Original Model | Solve for x: 3x + 7 = 25. | no | Okay, so I need to solve the equation 3x + 7 = 25. Let me think about how to approach this. I remember from algebra that the goal is to isolate x, which means getting x by itself on one side of the equation. First, I should probably get rid of that 7 that's being added to 3x. To do that, I can subtract 7 from both sides of the equation. That ... |
| Prune SWs Neurons | | yes | Okay, so I need to solve this equation: 3x + 7 = 25. Hmm, right. Let me think. So, first, maybe I should subtract 7 from both sides? Wait, no, that's not. Wait, no, that's not. Wait, no, that's not. Wait, no, that's not. Wait, no, that's not. Wait, no, that's not. Wait, no, that's not. Wait, no, that's not. Wait, no, that's not. Wait, no, that's not. Wait, no, that's not. Wait, no, that's not. Wait, no, that's not. Wait, no, that's not. ... |
| Random Pruning | | no | Okay, so I need to solve the equation 3x + 7 = 25. Let me think about how to approach this. I remember from algebra that the goal is to isolate x, which means getting x by itself on one side of the equation. First, I should probably get rid of that 7 that's being added to 3x. To do that, I can subtract 7 from both sides of the equation. That ... |

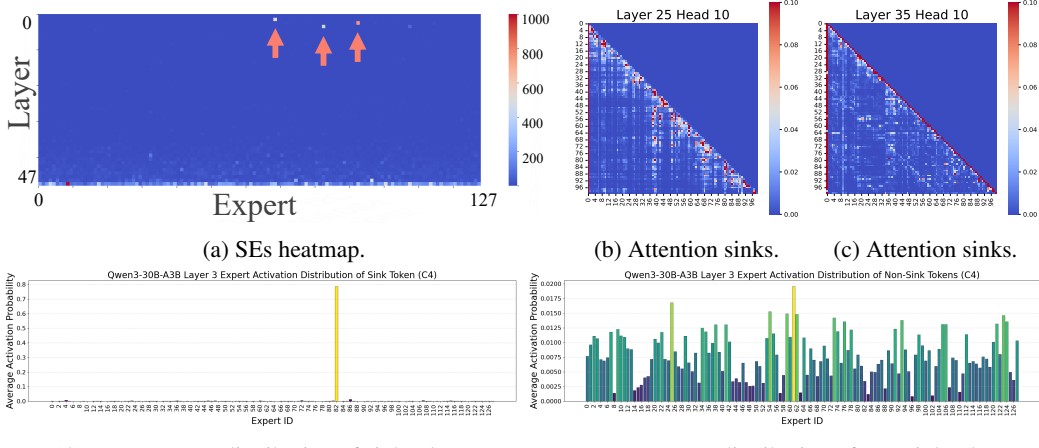

(a) SEs heatmap.

(b) Attention sinks.

(c) Attention sinks.

(d) Router scores distribution of sink token.

(e) Router scores distribution of non-sink tokens.

Figure 24: SE heatmaps, attention sink visualizations, and router score analyses for sink and non-sink tokens in Qwen3-30B-A3B on the C4 dataset. SEs are highlighted with arrows.

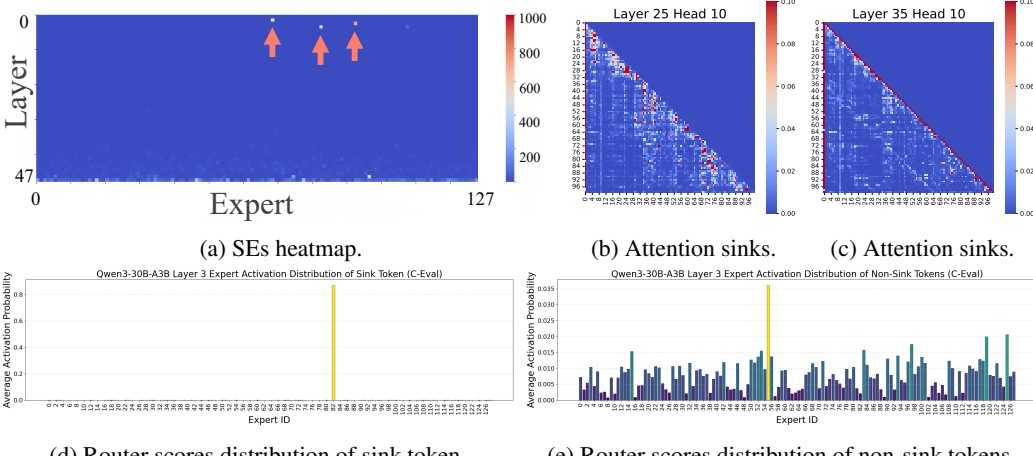

(a) SEs heatmap.

(b) Attention sinks.

(c) Attention sinks.

(d) Router scores distribution of sink token.

(e) Router scores distribution of non-sink tokens.

Figure 25: SE heatmaps, attention sink visualizations, and router score analyses for sink and non-sink tokens in Qwen3-30B-A3B based on the C-Eval dataset. SEs are highlighted with arrows.

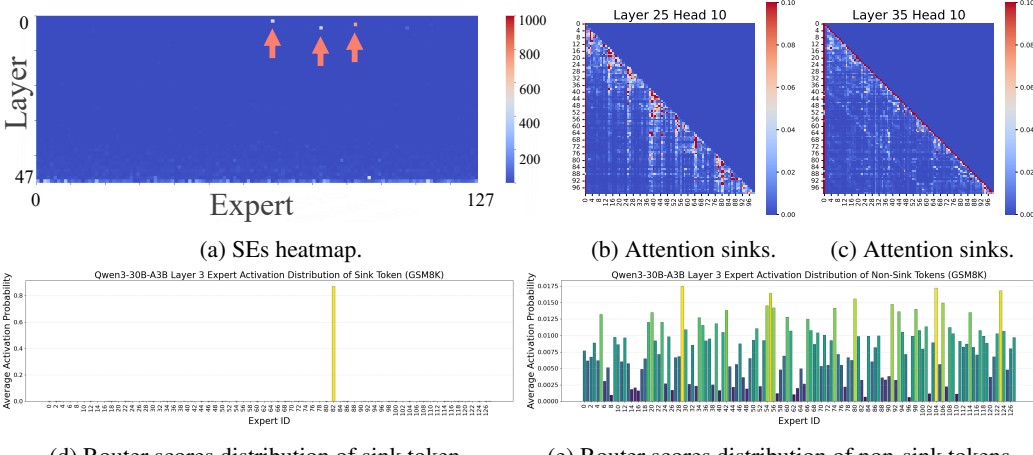

(a) SEs heatmap.

(b) Attention sinks.

(c) Attention sinks.

(d) Router scores distribution of sink token.

(e) Router scores distribution of non-sink tokens.

Figure 26: SE heatmaps, attention sink visualizations, and router score analyses for sink and non-sink tokens in Qwen3-30B-A3B based on the GSM8K dataset. SEs are highlighted with arrows.

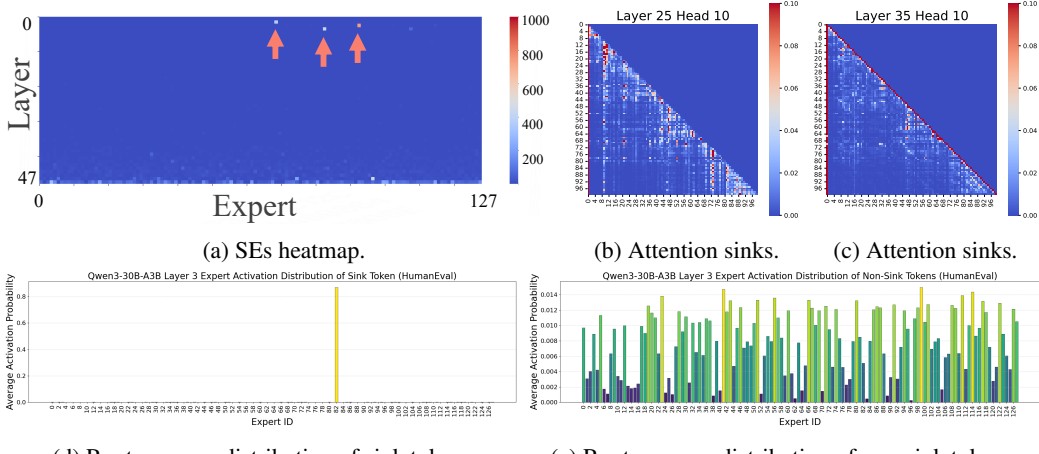

Figure 27: SE heatmaps, attention sink visualizations, and router score analyses for sink and non-sink tokens in Qwen3-30B-A3B based on the HumanEval dataset. SEs are highlighted with arrows.

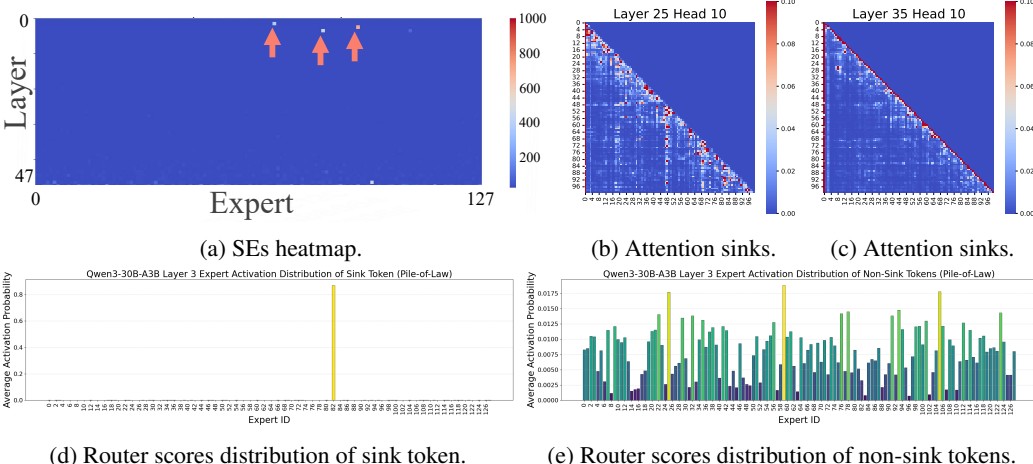

Figure 28: SE heatmaps, attention sink visualizations, and router score analyses for sink and non-sink tokens in Qwen3-30B-A3B based on the Pile-of-Law dataset. SEs are highlighted with arrows.

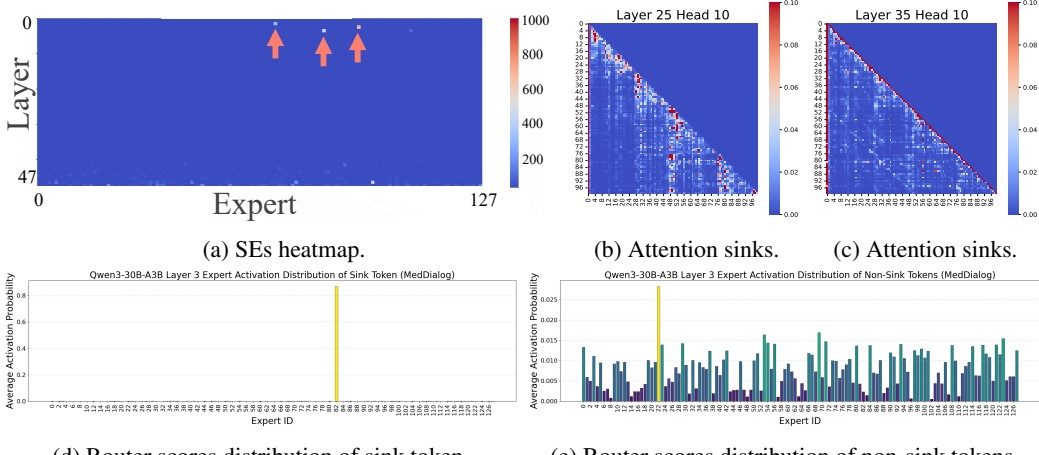

Figure 29: SE heatmaps, attention sink visualizations, and router score analyses for sink and non-sink tokens in Qwen3-30B-A3B based on the MedDialog dataset. SEs are highlighted with arrows.

