# OpenReview forum: "Unveiling Super Experts in Mixture-of-Experts Large Language Models"
_ICLR.cc/2026/Conference — ICLR 2026 Poster_

### Official Review · Reviewer_mM7h · 2025-10-28

**Soundness:** 3
**Presentation:** 3
**Contribution:** 3
**Rating:** 6
**Confidence:** 4

**Summary:**

This paper investigates “super experts", specialized experts within large Mixture-of-Experts language models that disproportionately influence performance. The authors identify such experts through gradient-based sensitivity analysis and routing statistics, arguing that a small subset of experts accounts for the majority of downstream quality.

**Strengths:**

[1] Introduces the super-expert hypothesis, offering a conceptual lens distinct from purely performance-based pruning. Connects findings to neural specialization, offering a biological analogy that enriches interpretability discussions.
[2]  Strong empirical evidence across diverse MoE models (e.g., Mixtral-8×7B, DeepSeek-R1) and tasks (MMLU, GSM8K, CodeEval).   Good balance between theoretical framing and empirical results.
[3] Clear motivation and well-structured narrative.

**Weaknesses:**

[1]  If “super experts” arise due to overexposure to specific domain tokens, the effect might reflect data imbalance rather than emergent specialization. The paper does not control for or analyze this possibility.
[2] Results focus on held-out benchmarks similar to pretraining data. It is unclear if retaining only super experts maintains out-of-distribution robustness.
[3]  There is little theoretical explanation for why certain experts become super experts.

**Questions:**

See weakness.

Are “super experts” persistent across training epochs, or do they shift over time?

It would be valuable to provide per-domain heatmaps showing which experts dominate specific token types or tasks.

---

> ### Author Response · Authors · 2025-11-21
> **Official Comment by Authors (Part I)**
>
> We sincerely appreciate the reviewer’s careful evaluation of our work. We are grateful for the constructive feedback, which both highlights the strengths of our study and raises insightful questions. We will make every effort to address the questions and clarify potential misunderstandings.
>
> **1. Response to Weakness [1] and Question [2]**
>
>  We recognize that our initial explanation may not have fully conveyed that Super Experts（SEs）do not arise from over-exposure to domain-specific tokens. The sink token shown in Figure 6 not only receives disproportionately large router scores but also consistently corresponds to the first token across all evaluated models, irrespective of the domain or input prompt. This sink token is content-agnostic and domain-independent, determined solely by positional index rather than any semantic or domain characteristic (References [1-3]). Therefore, the emergence of SEs is not driven by data imbalance or overexposure to specific domain tokens. Instead, it reflects an intrinsic and cross-domain stable property of MoE LLMs.
>
> Regarding the request for per-domain analyses, our submission already includes domain-level heatmaps. Appendix D presents expert activation patterns for Chinese, English, mathematics, and code, covering linguistically and semantically diverse domains. Across all these domains, the identified set of SEs remains fully consistent, further supporting the domain-invariant nature of SEs.
>  To provide a more complete and transparent analysis, we additionally include in the revised version a unified presentation of the following visualizations in the newly added Appendix J:
>
> - per-domain SEs heatmaps,
> - per-domain attention sink visualizations, and
> - per-domain router score distributions for both sink and non-sink tokens
>
>  As shown in Figure [23-26], the distribution of SEs remains consistent across these varied domains. Moreover, the sink tokens consistently appear as the first token. Their router scores on SEs are markedly larger, whereas the router scores of non-sink tokens remain relatively uniform across experts. These results reinforce that SEs reflect an intrinsic property of MoE LLMs rather than arising from over-exposure to domain-specific tokens.
>
> **2. Response to Weakness [2]**
>
>  Out-of-distribution (OoD) robustness is indeed a crucial aspect to consider. Building on our main findings and the clarifications provided in Response [1], we expect that SEs remain stable even in domains substantially different from the pretraining data, as their emergence is driven by the domain-independent sink-token mechanism rather than data-specific patterns.
>
>  To validate this, we conducted additional experiments on OoD domains (References [4-5]), which are also presented in the newly added Appendix J. As shown in Figure [27-28], the distribution of SEs remains consistent across these OoD datasets, reinforcing our claim that SEs reflect an intrinsic property of MoE LLMs and are not affected by OoD inputs.

---

> ### Author Response · Authors · 2025-11-21
> **Official Comment by Authors (Part II)**
>
> **3. Response to Question [1]**
>
>  Our findings indicate that once SEs emerge, they remain stable throughout subsequent training stages. As shown in Table 2，Figures 5 and Appendix D, the SEs identified in the base versions of Qwen3, DeepSeek, and Mixtral, as well as in their instruct variants obtained through full-model fine-tuning over multiple training epochs, are completely consistent.
>
> We additionally identified SEs in four officially open-sourced Qwen3-30B-A3B models at different training stages: Qwen3-30B-A3B-Base, Qwen3-30B-A3B, Qwen3-30B-A3B-Instruct-2507, and Qwen3-30B-A3B-Thinking-2507. The indices of the SEs, together with their activation heatmaps, are presented in the newly added Appendix K, showing consistent sets across all models and training stages. These results support the conclusion that SEs are persistent features of MoE LLMs.
>
> **4. Response to Weakness [3]**
>
> This is indeed a question that we are also highly interested in. However, developing a full theory would require a detailed analysis of the training dynamics of industrial-scale MoE LLMs, which is challenging under current computational constraints. In the present work, our contributions focus on post-training analysis. We provide the first systematic identification and comprehensive characterization of SEs. We thoroughly examine their distributions, intrinsic properties, and critical functional roles in driving systematic outliers, as well as their impact on post-training expert-level compression. These findings advance the understanding of the internal dynamics of MoE LLMs and help fill an important gap in the current literature.
>
> Nevertheless, this remains an important direction for future work. In our ongoing efforts, we are investigating the formation of SEs during large-scale MoE pretraining. In addition, we are exploring recent methods for mitigating attention sinks during pretraining (References [6–9]) in order to reduce the extreme imbalance caused by SEs. We consider this an important opportunity for future research on SEs.
>
> We sincerely thank the reviewer for raising these points. We hope that the clarifications provided adequately address the concerns, and we would greatly appreciate it if the rating could be reconsidered. Thank you once again for your time and effort!
>
> **References**
>
> [1] Efficient Streaming Language Models with Attention Sinks
>
> [2] When Attention Sink Emerges in Language Models: An Empirical View
>
> [3] Active-Dormant Attention Heads: Mechanistically Demystifying Extreme-Token Phenomena in LLMs
>
> [4] Pile of Law: Learning Responsible Data Filtering from the Law and a 256GB Open-Source Legal Dataset
>
> [5] MedDialog: Two Large-scale Medical Dialogue Datasets
>
> [6] Gated Attention for Large Language Models: Non-linearity, Sparsity, and Attention-Sink-Free
>
> [7] Softpick: No Attention Sink, No Massive Activations with Rectified Softmax
>
> [8] gpt-oss-120b & gpt-oss-20b Model Card
>
> [9] Qwen3-Next: Towards Ultimate Training & Inference Efficiency

---

### Official Review · Reviewer_Jn1W · 2025-10-28

**Soundness:** 3
**Presentation:** 3
**Contribution:** 3
**Rating:** 6
**Confidence:** 4

**Summary:**

The authors introduce the concept of Super Experts (SEs) in Mixture-of-Experts (MoE) Large Language Models (LLMs), identifying them as the main source of activation outliers, a phenomenon analogous to the massive activations (MA) seen in dense LLMs. These SEs are few in number, yet their pruning severely degrades performance. The extreme activation outliers causing this behavior are found to arise specifically from the `down_proj` operation within the expert layers. A series of experiments confirmed that SEs share key characteristics with MA, such as being model-specific, data-agnostic, and having a relationship with Attention Sink (AS) tokens. Furthermore, their distributions are noted to remain unaffected after post-training, and their impact was demonstrated across both reasoning and non-reasoning tasks.

**Strengths:**

This work is the first to analyze the source of massive activations (MA) in Mixture-of-Experts (MoE) LLMs, providing a quantitative definition for this source by introducing the concept of Super Experts (SEs). The paper is well-written and easy to follow, and its core claims are supported by extensive experimental evidence.

**Weaknesses:**

See questions below.

**Questions:**

1. Since Super Experts (SEs) and Massive Activations (MAs) are hypothesized to function as an implicit, input-independent bias term, one would expect them to be primarily localized in global components, such as the shared expert in models like DeepSeek-R1, which is universally activated by all tokens. Therefore, could the authors provide a detailed explanation as to why SEs appear in both normal (token-specific) and shared experts, as demonstrated by the distribution shown in Table 7?

2. The original research on Massive Activations (MAs) in dense LLMs demonstrated that their presence could be eliminated by introducing an explicit, input-independent bias term to the hidden states. Have the authors investigated whether implementing a comparable, explicit bias term technique in MoE models can eliminate the appearance of Super Experts (SEs) or significantly mitigate their performance-critical reliance on them?

3. A key contribution of this work is the distinction between activation outliers in MoE and dense LLMs. The observation that "Unlike dense LLMs, where such behavior typically occurs within a single layer, MoE models exhibit the progressive formation of systematic outliers by SEs across multiple layers." (illustrated in Figures 7 and 19) is particularly intriguing but lacks detailed discussion. Could the authors elaborate on this "progressive formation" phenomenon, providing a comparative analysis of why this multi-layer progression is necessary or emerges in the MoE architecture, while the analogous behavior in dense LLMs appears to be self-contained within a single layer?

4. Could the authors include an extra section that summarizes the similarities and differences between the activation outlier phenomena in dense LLMs and MoE models (maybe in the appendix)? This section should detail their shared characteristics, such as both being model-specific, data-agnostic, and having a relationship with Attention Sink (AS) tokens, while clearly highlighting key distinctions, such as the outlier source transitioning from a single-layer event in dense models to a multi-layer progressive formation in MoE architectures, and so on.

---

> ### Author Response · Authors · 2025-11-21
> **Official Comment by Authors (Part I)**
>
> We sincerely appreciate the reviewer’s careful evaluation of our work. We are grateful for the constructive feedback, which both highlights the strengths of our study and raises insightful questions. We will make every effort to address the questions and clarify potential misunderstandings.
>
> **1.Response to Question [1]**
>
>  Building on the concept of attention bias and the way it arises in MoE LLMs through Super Experts (SEs), we believe that SEs are not necessarily confined to global components (i.e., shared experts). In the following, we provide a detailed explanation by discussing the formulation process of attention bias in MoE LLMs.
>
> The overall process can be summarized as follows:
>
> **Step 1:** The sink token is strongly activated by Super Experts (SEs) in MoE layers (our work).
>
> **Step 2:** The sink token generates massive activations (MAs) in the hidden states (References [1–2]).
>
> **Step 3:** The sink token produces attention sinks (ASs) in the attention layers (Reference [3]).
>
> **Step 4:** These attention sinks induce attention biases on all subsequent tokens (Reference [1]).
>
>  In the crucial first step, in order to trigger the subsequent MAs and ASs, the sink token’s router logits need to be unusually skewed toward the SEs among the sparsely activated experts. In contrast, non-sink tokens are distributed more evenly due to pre-training load-balancing. This behavior is illustrated in Figures 6 and 13–18, which corroborates the above discussion.
>
>  Regarding whether SEs must necessarily appear in globally shared components, we believe that this is not required. First, the presence of routers allows the skewed routing scores themselves to induce a stable attention bias, in a manner analogous to globally shared structures. Second, the source of the attention bias only requires that the sink token (which is typically the first token (References [2])) is activated by SEs within the MoE layers. In contrast, shared experts activate all tokens indiscriminately.
>
> Accordingly, we hypothesize that having the router selectively and strongly activate only the sink token represents a more effective mechanism learned by the MoE LLMs.
>
> **2.Response to Question [2]**
>
>  This is indeed a topic of great interest to us. Our preliminary view is that introducing explicit attention bias has the potential to mitigate SEs, as attention bias can reduce MAs and ASs (References [1, 3–4]), and SEs, MAs, and ASs are all systematic phenomena within MoE LLMs (Section 5.1). For developing a complete theoretical understanding, it would require a detailed analysis of the training dynamics of industrial-scale MoE LLMs. In our ongoing efforts, we are exploring recent techniques for mitigating attention sinks during pretraining—including explicit attention bias and approaches proposed in References [5–9]—with the goal of reducing the extreme imbalance introduced by SEs. We believe this line of investigation offers a promising path toward better controlling SE behavior.
>
>  In the present work, our contributions focus on post-training analysis. We provide the first systematic identification and comprehensive characterization of SEs, thoroughly examining their distributions, intrinsic properties, and critical functional roles in driving systematic outliers, as well as their impact on post-training expert-level compression. These findings advance the understanding of the internal dynamics of MoE LLMs and address an important gap in the current literature.

---

> ### Author Response · Authors · 2025-11-21
> **Official Comment by Authors (Part II)**
>
> **3.Response to Question [3]&[4]**
>
>  As shown in Figures 3 and 4 and summarized in Table 2, the SEs in Qwen3 and DeepSeek originate from different layers, and the corresponding MAs accumulate progressively across layers. In contrast, in dense LLMs such as LLaMA-2 and LLaMA-3, MAs typically emerge within a single layer (References [1]). We have added Table 11 in Appendix L to summarize their shared characteristics and key differences.
>
>  While we acknowledge that a complete theoretical explanation for the progressive formation of SE-induced outliers across layers remains an open question and requires further investigation during MoE LLM pretraining, we offer the following plausible hypothesis. MoE LLMs operate under a fundamentally different mechanism from dense LLMs due to their sparsely activated MoE layers. Although the sink token typically receives high router scores, the load-balancing constraints imposed during pretraining prevent it from being routed to specific SEs with a 100% activation rate. As a result, the model naturally encourages a multi-layer progressive formation process, which may provide a more robust way to establish stable attention sinks.
>
>  Additionally, as noted in our Response 2, we are actively investigating the formation of SEs during pretraining, and understanding this multi-layer progression—both theoretically and empirically—remains a key focus of our ongoing work.
>
> We sincerely thank the reviewer for raising these points. We hope that the clarifications provided adequately address the concerns, and we would greatly appreciate it if the rating could be reconsidered. Thank you once again for your time and effort!
>
> **References**
>
> [1] Massive Activations in Large Language Models
>
> [2] Efficient Streaming Language Models with Attention Sinks
>
> [3] Systematic Outliers in Large Language Models
>
> [4] When Attention Sink Emerges in Language Models: An Empirical View
>
> [5] Quantizable Transformers: Removing Outliers by Helping Attention Heads Do Nothing
>
> [6] Gated Attention for Large Language Models: Non-linearity, Sparsity, and Attention-Sink-Free
>
> [7] Softpick: No Attention Sink, No Massive Activations with Rectified Softmax
>
> [8] gpt-oss-120b & gpt-oss-20b Model Card
>
> [9] Vision Transformers Need Registers

---

### Official Review · Reviewer_vNxZ · 2025-10-28

**Soundness:** 2
**Presentation:** 3
**Contribution:** 2
**Rating:** 4
**Confidence:** 4

**Summary:**

This paper investigates the heterogeneous importance of experts in Mixture-of-Experts (MoE) large language models. The authors report the discovery of a small, distinct subset of experts, which they term "Super Experts" (SEs). The central claim is that these SEs, despite being extremely rare (e.g., 3 out of 6,144 in Qwen3-30B-A3B), are fundamentally critical to the model's performance.

The authors provide strong empirical evidence showing that pruning just these few SEs causes a catastrophic collapse in model performance, including massive increases in perplexity and near-zero accuracy on reasoning tasks. In contrast, pruning a large number (e.g., 1000) of non-SEs has a negligible impact.

Mechanistically, the paper links SEs to the phenomenon of "massive activations" (MAs). It proposes that SEs are characterized by rare but extreme activation outliers in their down_proj output. These outliers propagate through residual connections, creating MAs in the hidden states. The paper further hypothesizes that these SE-generated MAs are the primary source of the systematic outlier mechanism in Transformers, and that their function is to create and maintain "attention sinks" (ASs). Pruning the SEs breaks this mechanism, causing the attention sink to collapse and, consequently, the model's performance to degrade, often resulting in repetitive and uninformative output. The authors also find that SE distribution is model-specific, data-agnostic, and stable through post-training.

**Strengths:**

1. **Interesting Empirical Finding**: The paper identifies a clear and striking phenomenon: a tiny, identifiable subset of experts is responsible for the vast majority of model stability. The empirical evidence, particularly the stark contrast between pruning SEs versus a large number of random experts (Fig. 1, Tables 3-5), is convincing.

2. **Plausible Mechanistic Explanation**: The paper provides a commendable in-depth analysis of why these experts are so critical. It successfully links the MoE structure to the known "massive activation" and "attention sink" phenomena in dense Transformers.

3. **Thorough Investigation**: The authors analyze this phenomenon across multiple models (Qwen, DeepSeek, Mixtral) and datasets, demonstrating that the presence of SEs is a consistent pattern. The analysis of router scores (Fig. 6), showing that sink tokens are preferentially routed to SEs, is a key piece of evidence for the proposed mechanism.

**Weaknesses:**

1. **Is this a new discovery or a restatement?** The paper's core finding is that SEs cause MAs, which in turn create attention sinks. The phenomenon and importance of massive activation are already known in the established literature. The paper's main contribution seems to be locating the source of MAs within specific experts in MoE models. In appendix H, the author also provides analysis on locating superweights within super experts, causing massive activation. It's unclear to me whether "Super Expert" is a new concept or just "the expert containing the MA neurons." The framing suggests a novel discovery, but it might be more of a localization and restatement of an existing concept

2. **"Super” experts could be misleading**: My **most significant concern** is the framing of the discovery. The term "Super Expert" implies these experts are qualitatively "better," "stronger," or has a superior capability. However, the paper's own evidence strongly suggests they are highly specialized components. Their function appears to maintain the attention sink. Their "importance" stems from the fact that the model's internal mechanism is critically dependent on them; pruning them removes attntion sink, causing collapse. This is different from removing the "smartest" expert. This naming convention is misleading and obscures the paper's core finding, which is about a specialized mechanism, not capability.

3. **Expert-level or neuron-level phenomena**: The paper frames this as an expert-level discovery. However, the mechanism is attributed to "extreme activation outliers" which, as investigated in Appendix H, can be traced to specific "Super Weights" (SWs) or neurons. This raises the question: is this an expert-level phenomenon, or are we simply observing the location of a few critical neurons (which cause massive activation)? The findings in Appendix H, where pruning just the neurons causes a similar collapse, suggest the latter. This point significantly muddies the central claim that this is about "experts" as a unit.

4. **Excluding final-layer activations is unconvincing**: The paper notes (primarily in Appendix C) that other experts, particularly in the final layers, also exhibit extreme activation outliers. These experts are excluded from "SEs" because they "do not contribute to the formation of MAs" and pruning them does not impact performance (Table 6). This reasoning feels post hoc and incomplete. In my opinion, it seems that removing MA in the final layer does not destroy the attention sink and thus does not destroy the performance. But this also corresponds to weakness 2; the findings in this paper should not be framed as identifying the super expert in the model.

**Questions:**

1. The analysis in Appendix H on "Super Weights" (SWs) is fascinating and, in my view, central to the paper's claim. It shows that pruning just a few neurons can replicate the catastrophic failure. To de-conflate the "expert" vs. "neuron" contributions, have the authors considered two complementary experiments? a) prune only the SWs identified in Appendix H, but leave the rest of the SEs' parameters intact. b) prune the entire SE except for the identified SWs. These experiments would clarify whether the model relies on the entire expert or just on these specific activation-generating channels.

2. Would the authors consider a more descriptive and less misleading name for this phenomenon to more accurately reflect the paper's findings?

3. Can the authors please clarify the definition of the layer set $L$ (layers responsible for MA formation) in Algorithm 1? How is this set determined a priori? How does this definition mechanistically exclude the final layers, even if they produce large outliers? Is the only reason for their exclusion the post-hoc empirical finding that pruning them does not hurt performance?

4. Could authors share insights on why the last layer produces more outliers in the expert output?

---

> ### Author Response · Authors · 2025-11-21
> **Official Comment by Authors (Part I)**
>
> We sincerely appreciate the reviewer’s careful evaluation of our work. We are grateful for the constructive feedback, which both highlights the strengths of our study and raises insightful questions. We will make every effort to address the questions and clarify potential misunderstandings.
>
> **1.Response to Weakness [1], Weakness [3] and Question [1]**
>
>  We understand that the core of the reviewer’s concerns lies in whether the proposed Super Experts (SEs) represent a genuinely expert-level mechanism or whether they are merely a reframing of neuron- or weight-level phenomena associated with Massive Activations (MAs). This is a valid and important point that deserves careful consideration. Below, we address this issue from three perspectives: Distinctive Features of SEs, Expert-Level Perspective with Practical Implications, and Significance of Our Contributions.
>
> **1.1  SEs Beyond Neuron-Level Effects**
>
>  While SEs are linked to neuron-level phenomena, they provide insights that cannot be reduced to mere neuron- or weight-level effects. Although MoE training emphasizes load balancing (Reference [1]), the router scores for the sink token are heavily skewed toward specific SEs, whereas the scores for non-sink tokens are distributed more evenly across all experts, as shown in Figure 6, 23-28. Defining SEs solely at the neuron level overlooks the critical role of the MoE routing mechanism. As a key component of MoE LLMs, the router determines which experts each token is assigned to and performs expert-level routing, serving as a central mechanism of MoE LLMs (Reference [1]). Accordingly, we consider SEs to constitute a expert-level mechanism, whose significance arises from the systematic process by which internal weights generate massive activations under the router’s specialized behavior.
>
> **1.2 Expert-Level Perspective with Practical Implications**
>
>  Although MoE LLMs provide strong scalability, they also incur substantial deployment costs. Structured, expert-level compression methods have attracted increasing attention (References [2–5]). Nevertheless, prior work lacks a thorough understanding of the heterogeneous importance among experts and overlooks the presence of exceptionally critical experts. In this context, our work not only identifies these highly influential experts but also systematically investigates the sources of their significance, as detailed in Section 5. Specifically, their importance arises from systematic outliers in Transformers under the router’s specialized behavior in MoE models. This phenomenon reflects an emergent property resulting from the interaction between Transformers and the MoE architecture. By uncovering SEs, our work clarifies their potential implications for expert-level compression and further offers insights for mitigating extreme expert imbalance during pretraining as well as improving the effectiveness of post-training expert-level compression.
>
> **1.3 Significance of Our Contributions**
>
>  State-of-the-art LLMs are increasingly adopting MoE architectures (References [1, 6–9]). However, prior studies on MAs and systematic outliers in Transformers have largely focused on dense LLMs (References [10–11]), making them somewhat outdated in the rapidly evolving LLM landscape. Before our work, existing understanding could not answer whether neuron-level outliers appear across all experts or are instead concentrated in a few specialized experts, nor how the router in MoE models interacts with Transformer to produce these systematic outliers. To the best of our knowledge, our work is the first to systematically characterize the mechanism of exceptionally critical experts in MoE LLMs and to establish a clear explanation linking these experts to systematic outliers in Transformers, thereby resolving a key missing piece in current understanding.
>
> **1.4 Response to Question [1]**
>
>  Regarding Question 1, we believe that our analyses in Figure 7 and Figure 21 already demonstrate that the importance of SEs stems from their role in constructing systematic outliers. However, as discussed in our response in “SEs Beyond Neuron-Level Effects”, this weight-level criticality cannot be decoupled from the expert-level routing behavior. Thus, isolating SWs from the expert-level context does not fully capture the underlying mechanism.

---

> ### Author Response · Authors · 2025-11-21
> **Official Comment by Authors (Part II)**
>
> **2.Response to Weakness [2] and Question [2]**
>
>  We fully understand how this terminology may lead to alternative interpretations, and we sincerely appreciate the reviewer for highlighting this potential ambiguity. We considered several alternatives such as “Critical Experts” and “Outlier-Generating Experts,” and ultimately chose the term super experts. Our intention was not to suggest that these experts possess superior capability or are “better” in any qualitative sense. Instead, the term “super” was selected to underscore their exceptional functional importance—specifically, their disproportionate contribution to inducing structured MAs and attention sinks that are essential for maintaining stable model behavior. In this context, “super” refers to a high degree of importance, rather than enhanced capability or intelligence. These experts are highly specialized components whose significance arises from their structural role within the model’s internal mechanism, rather than from being intrinsically “stronger” experts.
>
> **3.Response to Weakness [4] and Question [4]**
>
>  Outlier experts in the final layers are excluded from SEs not merely because pruning them does not affect model performance. The MAs emerge in earlier layers and remain stable across most intermediate layers, where they contribute to the formation of attention sinks (References[10-11]). Therefore, these final-layer outliers do not participate in the systematic outlier mechanism that defines the importance of SEs. A key piece of evidence is that the router scores in the final layer are relatively evenly distributed across all experts for both sink and non-sink tokens, as shown in the Figure 13 in newly added Appendix M. In contrast, sink-token router logits are strongly skewed toward SEs, indicating that these final-layer outlier experts are fundamentally different from SEs. Pruning experiments on these final-layer outlier experts just further confirm that they do not possess the critical functional importance of SEs.
>
>  While we do not yet have a definitive explanation for why the final layer exhibits more extreme activations, a plausible factor is the role of the final layer in output distribution. Activations here primarily influence the final output logits, and final-layer experts may amplify specific patterns or tokens, producing high-magnitude activations that appear as numerical outliers without directly contributing to the formation of attention sinks.
>
> **4.Response to Question [3]**
>
>  We thank the reviewer for raising this question. The layer set $L$ is not determined based on post-hoc empirical observations. Rather, it is established through a lightweight pre-calibration process. The output activation distributions of the down-projection layers are highly stable across domains and input prompts, as illustrated by the SE heatmaps for Qwen3, DeepSeek, and Mixtral in Figure 24-29. Consequently, a single forward pass during this calibration is sufficient to identify, a priori, the layers responsible for generating SEs.
>
> We sincerely thank the reviewer for raising these points. We hope that the clarifications provided adequately address the concerns, and we would greatly appreciate it if the rating could be reconsidered. Thank you once again for your time and effort!
>
> **Reference:**
>
> [1] DeepSeek-V3 Technical Report
>
> [2] Mixture Compressor for Mixture-of-Experts LLMs Gains More
>
> [3] MxMoE: Mixed-precision Quantization for MoE with Accuracy and Performance Co-Design
>
> [4] Not All Experts are Equal: Efficient Expert Pruning and Skipping for Mixture-of-Experts Large Language Models
>
> [5] MoE-I^2: Compressing Mixture of Experts Models through Inter-Expert Pruning and Intra-Expert Low-Rank Decomposition
>
> [6] Qwen3 Technical Report
>
> [7] Kimi K2: Open Agentic Intelligence
>
> [8] LongCat-Flash Technical Report
>
> [9] gpt-oss-120b & gpt-oss-20b Model Card
>
> [10] Massive Activations in Large Language Models
>
> [11] Systematic Outliers in Large Language Models

---

> > ### Comment · Reviewer_vNxZ · 2025-11-24
> > **Response to authors**
> >
> > Thanks for the detailed response. While I appreciate the evidence regarding the router's behavior (i.e., the router consistently directing sink tokens to these specific experts), **I do not find that this fully resolves my core concern** regarding whether this is an "Expert-level" or "Neuron-level" phenomenon.
> >
> > My reasons are as follows:
> >
> > 1. Framing this at the **neuron level does not conflict with studying its interaction with expert-level routing**. We can acknowledge that specific neurons (Super Weights) are the root cause of the Massive Activations (MAs) while simultaneously investigating how these particular neurons influence or correlate with the router's decisions.
> > 2. My primary concern with framing this as "Expert-level" is that it implies the MA formation and the Attention Sink are the **collective impact** of the expert's parameters as a whole unit. If the phenomenon is driven solely by a handful of specific neurons within that expert, attributing the mechanism to the Expert is scientifically imprecise.
> >
> > If pruning only the identified Super Weights (SWs) results in a performance collapse comparable to pruning the entire Super Expert (SE), the mechanism is functionally located at the neuron level, not the expert level. Therefore, I strongly suggest the authors perform the requested ablation experiment to validate the framing at the expert level: **Compare the performance drop between (a) pruning only the identified Super Weights within the SEs and (b) pruning the entire SEs. (c) [optional] pruning a similar amount of non-SE random neurons**.
> >
> > I will raise my score if the authors clearly demonstrate this performance gap because it naturally addresses my primary concern. If the results are similar, I believe the paper would be significantly strengthened by reframing the discovery of localizing superwights in the MoE model rather than claiming the existence of "Super Experts".

---

> > > ### Author Response · Authors · 2025-11-27
> > > **Response to Reviewer vNxZ (Part I)**
> > >
> > > Thank you very much for your timely response and for your willingness to engage in this discussion. Nevertheless, based on our understanding and the new evidence we have gathered, **we maintain that Super Experts (SEs) in MoE LLMs constitute an expert-level mechanism**. In short, while Super Weights (SWs) highlight the importance of weight-level contributions, this does not justify interpreting SEs as solely a neuron-level phenomenon. In MoE LLMs, such an interpretation could overlook several critical insights that become apparent when the analysis is conducted from an expert-level perspective.
> > >
> > > To address your concern as thoroughly as possible, we have included additional analyses, new experiments, and detailed interpretations, which are presented below:
> > >
> > > **1. Response to your reasons [2]:**
> > >
> > > We believe that defining “expert-level” as necessarily the collective impact of all expert parameters on Massive Activation (MA) and Attention Sink (AS) formation is not entirely accurate.
> > >
> > > **First**, if a mechanism exhibits functional sparsity, it does not require the involvement of all parameters to achieve its purpose. Specifically, there can exist a highly consistent subset of experts (SEs) that are consistently specialized to perform a single function, i.e., MA and AS formation, where only a subset of internal weights is critical for this function, while the remaining weights are redundant or contribute merely as noise.
> > >
> > > Such a case should still be considered an expert-level mechanism, because the expert internally exhibits systematic coordination between critical and non-critical weights. Focusing only on the critical weights and interpreting the mechanism as neuron-level would overlook the “dark matter” of the expert, meaning the non-critical weights. **We have empirically validated this hypothesis and provide a detailed analysis in Response 3.**
> > >
> > > In fact, sparsity is common in Transformers. For example, attention patterns are highly sparse, with a small number of tokens receiving the majority of attention scores. This sparsity can be leveraged to reduce computation and accelerate attention operations (References[1-3]).
> > >
> > > **Second**, we emphasize another key aspect of interpreting SEs as an expert-level mechanism. SWs appear only in a small subset of experts. This selective presence establishes a clear distinction at the expert level. If all experts consistently displayed SWs, with no differences across experts, then interpreting the phenomenon purely as neuron-level would indeed be more appropriate. As part of our contributions, we show that in MoE LLMs, SEs consistently arise in only a very small number of experts—independent of input, data domain, or training dynamics. Therefore, neuron-level interpretation would necessarily need to be grounded in this highly specific expert-level variation.
> > >
> > > **2. Response to your reasons [1]:**
> > >
> > > We would like to respond by highlighting the following two points.
> > >
> > > **First**, it is crucial to interpret the mechanism within the MoE LLM framework, as our study focuses on MoE LLMs. While every mechanism can ultimately be attributed to the actions of neurons within experts, reducing the analysis solely to the neuron level risks overlooking the unique characteristics of MoE models. In MoE LLMs, neurons operate within experts that are first selected by the router. As part of our contributions, we show that the router consistently directs sink tokens to SEs, which represents a fundamentally expert-level behavior. This provides a basis for further investigation of finer-grained neuron-level attributions.
> > >
> > > **Second**, the router assigns tokens to different experts based on input distributions, and its ability to select experts is established during pretraining. During the forward pass, SWs emerge only after the router’s selection in a given layer. Therefore, SWs cannot influence the router’s choices. When processing sink tokens, the router consistently skews assignments toward SEs, and this capability is acquired during pretraining, demonstrating that SEs are an intrinsic property of MoE LLMs.

---

> > > ### Author Response · Authors · 2025-11-27
> > > **Response to Reviewer vNxZ (Part II)**
> > >
> > > **3. Response to the Request for Additional Experiments:**
> > >
> > > To provide as comprehensive an explanation as possible, we conducted the experiments suggested by the reviewer, as well as additional related experiments. **However, as noted in Response 1, based on an in-depth experimental analysis, we hold a different interpretation of the results.** The conclusions are summarized as follows:
> > >
> > > - **SEs exhibit highly sparse functionality, with internal weights falling into two extreme states: critically important and largely insignificant.** As shown in Table 1, pruning both SEs and SWs simultaneously results in substantial model collapse, indicating that SWs capture the weight-level criticality within SEs. Moreover, pruning the majority of non-SW weights within SEs has a negligible impact on model performance.
> > >
> > > - **SEs should be considered an expert-level mechanism because the expert internally exhibits systematic coordination between critical and non-critical weights.** This pronounced disparity in weight importance within experts contrasts sharply with dense LLMs, where non-SW weights remain highly significant. As shown in Table 2, pruning non-SW weights in dense LLMs still results in model collapse.
> > >
> > > - Integrating the observations from points 1 and 2, along with our analysis in the manuscript, we can conclude that in MoE LLMs, **SEs are specialized experts primarily responsible for the single function of generating MA and AS. They are few in number, highly consistent, and strongly activated for sink tokens, with this function being highly sparse.** This phenomenon arises from the interaction between the MoE architecture and the systematic outlier characteristics of Transformers, and, as one of our contributions, it is identified and reported for the first time.
> > >
> > > | Table 1: Qwen3-30B-A3B| WikiText-2 PPL| C4 PPL| Avg.  | ARC-c   | ARC-e | BoolQ | HellaSwag | MMLU  | PIQA  | WinoGrande |
> > > |-------------------------------------------|---------|--------|-------|-------|-------|-------|-----------|-------|-------|------------|
> > > | original                                  | 8.71    | 13.06  | 72.60 | 52.65 | 79.50 | 88.72 | 59.63     | 77.82 | 79.33 | 70.56      |
> > > | Prune SEs  | 59.70   | 74.07  | 55.64 | 46.08 | 76.05 | 70.73 | 39.31     | 56.03 | 29.80 | 71.51      |
> > > | Prune SWs (num=3)                         | 48.11   | 62.13  | 58.07 | 42.66 | 73.36 | 71.04 | 35.78     | 52.65 | 71.33 | 59.67      |
> > > | Prune all non-SW weights (num=2048×768×3−3) | 8.69    | 13.07  | 72.33 | 51.79 | 78.45 | 88.23 | 59.57     | 77.66 | 79.71 | 70.93|
> > >
> > > | Table 2: Llama-3-8B| WikiText-2 PPL| C4 PPL| Avg.  | ARC-c   | ARC-e | BoolQ | HellaSwag | MMLU  | PIQA  | WinoGrande |
> > > |-------------------------------------------|----------------|-----------|-------|-------|-------|-------|-----------|-------|-------|------------|
> > > | original | 6.14| 9.24| 69.48 | 50.34 | 80.09 | 81.35 | 60.18| 62.18 | 79.60 | 72.61|
> > > | Prune Layer 1 down_proj       | 320.88    | 263.15    | 36.03 | 18.43 | 29.25 | 48.07 | 27.52| 24.04 | 54.57 | 50.36|
> > > | Prune SWs (num=3)| 241832.09 | 286665.28 | 35.65 | 20.82 | 24.96 | 54.53 | 25.30| 22.95 | 52.50 | 48.46      |
> > > | Prune all non-SW weights (num=4096×14336−3) | 320.88| 263.15| 36.03 | 18.43 | 29.25 | 48.07 | 27.52| 24.04 | 54.57 | 50.36|

---

> > > ### Author Response · Authors · 2025-11-27
> > > **Response to Reviewer vNxZ (Part III)**
> > >
> > > **4. Discussion from a Broader View**
> > >
> > > In this section, we kindly invite the reviewer to consider the question of expert-level versus neuron-level from a broader perspective.
> > >
> > > - Mainstream LLM architectures continue to evolve, and we should adopt a developmental view when examining the interplay between the MoE structure and the intrinsic outlier characteristics of Transformers. Expert-level and neuron-level phenomena may simply reflect different interpretative perspectives, and in MoE LLMs, they coexist and influence each other. SWs explain how the phenomenon arises, while SEs indicate where it consistently emerges. By adopting a developmental perspective, we can interpret the inherent patterns of traditional Transformers through the lens of the new characteristics of MoE architectures, allowing us to more clearly recognize and embrace emerging changes.
> > >
> > > - Beyond this question, our work still makes a critical contribution by addressing a significant gap in the understanding of rapidly evolving LLMs, as highlighted in the sections **"Significance of Our Contributions"** and the **"Expert-Level Perspective with Practical Implications"** in previous responses. Prior literature on systematic outliers has not examined MoE architectures in detail. Our study provides the first in-depth analysis of MoE LLMs in this context, offering novel and independent insights that are not accessible through studies of dense LLMs.
> > >
> > > - From the perspective of implementing efficient structured compression (i.e., expert-level compression), even if a few critical weights are identified while the remainder are less significant, it is not feasible to selectively preserve only those weights during compression. In contrast, an expert-level analysis of importance can directly guide the active research community in pratical expert-level compression, addressing existing gaps in understanding and filling critical knowledge voids.
> > >
> > > Finally, while our perspective and responses may not fully align with your view or satisfy your criteria for raising score, we remain committed to our interpretation of SEs as reflecting effects beyond the neuron level. We sincerely appreciate and have greatly valued the opportunity to engage in this constructive discussion. We will update the manuscript to highlight our key insights regarding whether SEs should be considered expert-level or neuron-level.
> > >
> > > In any case, we are deeply grateful for your feedback, and we sincerely hope you might consider our perspective and contributions from a fresher and more open viewpoint, and, if possible, take into account a more appropriate rating.
> > >
> > > **References**
> > >
> > > [1] Native Sparse Attention: Hardware-Aligned and Natively Trainable Sparse Attention
> > >
> > > [2] MInference 1.0: Accelerating Pre-filling for Long-Context LLMs via Dynamic Sparse Attention
> > >
> > > [3] SpargeAttention: Accurate and Training-free Sparse Attention Accelerating Any Model Inference

---

### Official Review · Reviewer_r91y · 2025-10-31

**Soundness:** 3
**Presentation:** 3
**Contribution:** 3
**Rating:** 6
**Confidence:** 3

**Summary:**

This work identifies a rare subset of Super Experts in MoE LLMs, which generate extreme activation outliers that propagate through the model and trigger massive hidden state activations. The study further reveals that SEs serve as the primary source of systematic outliers by strongly activating attention sink tokens; compressing them severely disrupts attention sinks, ultimately causing repetitive and uninformative model outputs.

**Strengths:**

1. This paper presents the first rigorous identification and mechanistic explanation of "Super Experts" (SEs)—the primary source of systematic outlier phenomena, whose removal severely impairs performance, particularly in reasoning tasks.

2. By linking expert-level routing dynamics to Transformer-wide outlier mechanisms in MoE models, this work offers a principled account of known empirical phenomena and reveals critical vulnerabilities in current expert compression methods.

**Weaknesses:**

The identification of SEs heavily relies on an empirical threshold defined in Equation (6), where experts are selected based on exceeding the 99.5-percentile activation magnitude. This criterion appears rather extreme and ad-hoc, raising concerns that the observed stability of SEs may stem from the definition itself rather than reflecting an intrinsic property of the model.

**Questions:**

see weakness-1

---

> ### Author Response · Authors · 2025-11-21
>
> We sincerely appreciate the reviewer’s careful evaluation of our work. We are grateful for the constructive feedback, which both highlights the strengths of our study and raises insightful questions. We will make every effort to address the questions and clarify potential misunderstandings.
>
> **Response to Weaknesses and Questions**
>
>  We acknowledge that any threshold-based criterion requires careful validation to ensure that the observed phenomena reflect intrinsic model properties rather than methodological artifacts. Our central claim is that SEs represent a fundamental characteristic of MoE LLMs, while the threshold in Equation (6) provides a practical and principled mechanism for their identification. While a brief discussion is provided here, additional experiments are presented in the newly added Appendix I of the manuscript.
>
> **1. Scientific Rationale**
>
>  The identification of SEs is motivated by the heavy-tailed nature of extreme activations in the output of the down-projection layer from each expert. While most experts produce relatively modest activations, a small subset consistently generates extreme activations (as illustrated in Figure 5 and Appendix D), forming a distinct long tail that is naturally present in MoE LLMs. To mathematically validate this observation, we analyze the tail of the distribution using standard heavy-tail analysis techniques, including the complementary cumulative distribution function (CCDF) and power-law fitting, which allow us to quantify the tail behavior. As shown in Figure 11, the tail can be well-approximated by a power-law model, indicating that these extreme activations reflect inherent characteristics of the MoE LLMs rather than random noise.
>  This long-tail behavior highlights that only a few experts dominate the extreme activations, creating a clear separation from the majority. By focusing on this naturally occurring tail, threshold-based methods provide a practical and principled approach for identifying the small, statistically significant subset of experts responsible for these extreme activations.
>
> **2. Robustness of Super Experts Identification**
>
>  To assess the robustness of our identification procedure, we conduct a sensitivity analysis over a reasonable range of threshold values across different MoE LLMs and input domains. As shown in Table 9, the results demonstrate that SE identification remains consistent across these thresholds. As a result, the presence of SEs reflects an intrinsic property of MoE LLMs rather than an artifact of any specific threshold choice.
>
> We have revised the manuscript to:
> - Clarify the rationale for the threshold, highlighting its basis in the heavy-tailed distribution of expert activations.
> - Include results from a threshold-sensitivity analysis.
> - Emphasize that the stability of SEs reflects an intrinsic property of the MoE LLMs, with the threshold serving as a practical mechanism for their identification.
>
> We sincerely thank the reviewer for raising this point. We hope that the clarifications provided adequately address the concern, and we would greatly appreciate it if the rating could be reconsidered. Thank you once again for your time and effort!

---

### Author Response · Authors · 2025-11-27
**Kind Request for Feedback**

Dear Reviewers r91y, Jn1W and mM7h:

Since the discussion period is already halfway through, we kindly hope to receive your feedback. If there are any further questions or concerns, please feel free to point them out. We hope that the clarifications provided adequately address your concerns, and we would be very grateful if you could reconsider the rating in light of our responses.

Thank you once again for your time and effort!

---

### Author Response · Authors · 2025-11-30
**Summary of Prior Discussions**

Dear Area Chair,

We note the recent reviewer-leak incident on OpenReview, and we sincerely appreciate your time and effort in contributing to the ICLR review process. To assist your evaluation, we provide below a summary of the previous discussions for your convenience.

In short, we have provided comprehensive and systematic responses to each of the reviewers’ comments. However, due to the shortened discussion period, we received a response from only one reviewer, and no score changes were submitted.

***Reviewer r91y*** raised concerns regarding the threshold-based identification of Super Experts (SEs) and whether SE reflect an intrinsic property of the model. In response, we validated the heavy-tailed distribution of expert output activations and conducted threshold sensitivity experiments, demonstrating that the stability of SEs indeed reflects an intrinsic property of MoE LLMs, with the threshold serving as a practical identification mechanism. We believe these explanations address the reviewer’s concerns, and no further feedback was received.

 ***Reviewer vNxZ*** raised concerns regarding whether SEs in MoE LLMs reflect an expert-level or weight-level mechanism. We thank the reviewer for their attention and response. We engaged in a discussion, and our latest response has not yet received feedback. We believe the following key evidence indicates that SEs should be considered at the expert level, and that interpreting SEs solely as a neuron-level phenomenon is insufficient:

- Router scores for sink tokens are strongly skewed toward SEs, representing an expert-level routing behavior.

- SEs consistently arise in only a very small subset of experts, independent of input, data domain, or training dynamics. Therefore, a neuron-level interpretation would necessarily need to account for this highly specific expert-level routing.

- Importantly, our new experiments in the discussion show that SEs exhibit systematic coordination between critical and non-critical weights, further supporting the interpretation of SEs as an expert-level mechanism.

***Reviewer Jn1W*** raised questions regarding why SEs are not primarily localized in global components and whether the explicit bias term in MoE models can eliminate the appearance of SEs. We have responded to all of these questions and believe that most of the concerns have been adequately addressed, and no further feedback has been received.

***Reviewer mM7h***  suggested that SEs arise due to overexposure to specific domain tokens, which may involve some misunderstanding. In the revised version, we additionally include a unified presentation of per-domain visualizations and analyses, including out-of-distribution domains. These results reinforce that SEs reflect an intrinsic property of MoE LLMs rather than arising from overexposure to domain-specific tokens. We believe these explanations address the reviewer’s concerns, and no further feedback was received.

In closing, we sincerely thank you once again for your time and effort, and we hope that our summary provides you with a more complete understanding of our work.

---

### Meta-Review · Area_Chair_9c14 · 2026-01-13

**Summary:**

The paper discusses an intriguing phenomenon: fewer than 0.5% of experts in MoE language models are disproportionately critical for model function, with removing just three experts from Qwen3-30B-A3B causing perplexity to jump from 8.7 to 59.86. I find this observation valuable because it has direct implications for MoE compression and pruning research. The power-law analysis confirming the heavy-tailed structure underlying the identification criterion addresses concerns about arbitrary thresholds, and the validation across six domains including out-of-distribution data strengthens the generalizability of the findings. While one reviewer argues this may be a neuron-level rather than expert-level phenomenon, I consider the core observations well-supported and useful to the community regardless.

**Reviewer Concerns:**

The finding that a tiny subset of experts can catastrophically affect model performance has direct implications for MoE compression and pruning. The reviewers questioned whether the identification threshold was arbitrary, but power-law analysis confirms the heavy-tailed structure underlying the criterion. The domain dependence concerns were addressed through validation across six domains including out-of-distribution data. It should be noted that one reviewer argues this is a neuron-level phenomenon rather than expert-level, and that debate remains open. The core observations are well-supported and useful to the community regardless of how one frames the mechanism.

**Reviewer Scores:**

Scores were 6, 6, 6, and 4. The dissenting reviewer wants a different conceptual framing but does not dispute the experiments. Three out of four support acceptance.

---

### Decision · Program_Chairs · 2026-01-26

Accept (Poster)